neuroscience

blood-brain barrier, loud music, mechanisms of music-induced opening of the blood-brain barrier

**Author for correspondence:**
O. Semyachkina-Glushkovskaya
e-mail: glushkovskaya@mail.ru

# Phenomenon of music-induced opening of the blood-brain barrier in healthy mice

O. Semyachkina-Glushkovskaya[1,2], A. Esmat[2], D. Bragin[3,4], O. Bragina[3], A. A. Shirokov[1,5], N. Navolokin[1,6], Y. Yang[7], A. Abdurashitov[2], A. Khorovodov[2], A. Terskov[2], M. Klimova[2], A. Mamedova[2], I. Fedosov[2], V. Tuchin[2,8,9] and J. Kurths[1,2,10]

[1]Department of Physics, Humboldt University, Newtonstrasse 15, 12489 Berlin, Germany
[2]Department of Biology, Saratov State University, Astrakhanskaya Strasse 83, Saratov 410012, Russia
[3]Lovelace Biomedical Research Institute, Albuquerque, NM 87108, USA
[4]Department of Neurosurgery, University of New Mexico School of Medicine, Albuquerque, NM 87131, USA
[5]Institute of Biochemistry and Physiology of Plants and Microorganisms, Russian Academy of Sciences, Prospekt Entuziastov 13, Saratov 410049, Russian Federation
[6]Department of Anatomy, Saratov State Medical University, Bolshaya Kazachaya Strasse 112, Saratov 410012, Russia
[7]College of Pharmacy, University of New Mexico, Albuquerque, NM 87131, USA
[8]Laboratory of Biophotonics, Tomsk State University, 36 Lenin's Ave., Tomsk 634050, Russia
[9]Institute of Precision Mechanics and Control of RAS, Rabochaya Strasse 24, Saratov 410028, Russia
[10]Potsdam Institute for Climate Impact Research, Telegrafenberg A31, 14473 Potsdam, Germany

OS-G, 0000-0001-6753-7513

Music plays a more important role in our life than just being an entertainment. For example, it can be used as an anti-anxiety therapy of human and animals. However, the unsafe listening of loud music triggers hearing loss in millions of young people and professional musicians (rock, jazz and symphony orchestra) owing to exposure to damaging sound levels using personal audio devices or at noisy entertainment venues including nightclubs, discotheques, bars and concerts. Therefore, it is important to understand how loud music affects us. In this pioneering study on healthy mice, we discover that loud rock music below the safety threshold causes opening of the blood-brain barrier (OBBB), which plays a vital role in protecting the brain from viruses, bacteria and toxins. We clearly demonstrate that listening to loud music during 2 h in an intermittent adaptive regime is accompanied by delayed (1 h after music exposure) and short-lasting to (during 1–4 h) OBBB to low and high molecular weight compounds without cochlear and brain impairments. We present the systemic and molecular mechanisms responsible for music-induced OBBB. Finally, a revision of our traditional knowledge about the BBB nature and the novel strategies in optimizing of sound-mediated methods for brain drug delivery are discussed.

## 1. Introduction

Sounds, like music and noise, are an integral part of our life, and music is an important aspect of sound. We listen to various music such as classical, popular or rock on audio players, radio and TV, or during concerts. Whatever the type, music comprises what are known as notes, which are tones of sounds. Thus, the fundamental aspect of music is based on the concept of sound vibration, which owing to deep penetration into the brain and body affect individuals' mood and emotions positively or negatively at both the behavioural and neuronal level [1,2]. The study of music effects on the brain functions have significantly advanced in the last 30 years [3]. Basic and clinical findings suggest that listening to music involves many cognitive components with distinct brain substrates [4]. However, little is known about how music affects us and what are the mechanisms underlying this phenomenon. The World Health Organization estimated that 1.1 billion

teenagers and young adults are at risk of developing noise-induced hearing loss (NIHL) owing to the unsafe listening of loud music using personal audio devices such as smartphones and MP3 players and exposure to damaging levels of sound at noisy entertainment venues including nightclubs, discotheques, bars, pubs and sporting events [5]. For example, sounds at rock concerts routinely reach levels above 100 dB [6–8], which are considered unsafe for any unprotected exposures exceeding 15 min [9,10]. Rock, jazz and symphony orchestra musicians have been found to be at a significant risk of music-based NIHL [11–14].

The blood-labyrinth barrier (BLB) leakage is an important mechanism of NIHL [15]. Anatomically and functionally, BLB is similar to the blood-brain barrier (BBB) and comprises the endothelial cells in the microvasculature, the tight and adherens junctions, the pericytes and the basement membrane controlling the vascular permeability [15,16]. Loud sound causes a dramatic change in the strial cochlear–vascular unit [17] and an increase in BLB permeability leads to a number of NIHL [15,18].

Because the nature of BLB and BBB is similar and they are two keyplayers controlling the vascular permeability [15,19], it is expected that loud sound will increase the permeability of both the BLB and BBB. However, there is no information on how loud sound affects the BBB integrity.

In this *in vivo* and *ex vivo* experimental study on the healthy mice, we tested our hypothesis that loud music below the safety threshold can cause opening of the BBB (OBBB). We analysed the systemic and molecular mechanisms underlying this phenomenon and discussed advantages and disadvantages of music-induced OBBB.

## 2. Material and methods

The main material and methods are in the electronic supplementary material.

## 3. Results

### (a) The window of music-induced opening of the blood-brain barrier

In the first step, we determined the optimal duration/intensity of music exposure for OBBB as well as the time window of OBBB after listening to loud music (the design of our experiments is presented in the electronic supplementary material, figures S1–S3). Song of the Scorpions 'Still Loving You' was administered for periods of up to 0.25 (continues mode), 1 h and 2 h in an intermittent mode: 60 s sound and 60 s pause, at intensities ranging up to 70 dB (moderate), 90 dB (loud) and 100 dB (very loud) (https://www.asha.org/public/hearing/Noise/; http://www.industrialnoisecontrol.com/comparative-noise-examples.htm). Using the *in vivo* real-time fluorescent microscopy [20] in awake behaviour mice and optically cleared skull [21], we determined that a stimulus period of 2 h at 90–100 dB produced a robust increase in the BBB permeability to the Evans Blue Albumin Complex (EBAC, 68.5 kDa) that was observed as bright intensity around the cerebral capillaries (figure 1b). No changes in the BBB permeability were found in intact mice (no music, figure 1a) or when the stimulus was 70 dB or shorter times of music exposure (0.25 h and 1 h) was used. This *in vivo* technique also

revealed the delayed OBBB that was only 1 h after music effects but not immediately or 15/30 min after music-off that also we confirmed in *ex vivo* experiments using a spectrofluorometric quantitative assay of EBAC extravasation (electronic supplementary material, table S1).

These *ex vivo* findings also showed music-induced OBBB in 11 brain regions (electronic supplementary material, figure S3). The recovery of BBB was individual in each mouse with the high variability of units in the groups and wide recovery time range; however, there was no permeability of BBB to EBAC 4 h after music exposure in all mice (electronic supplementary material, table S1). There was no OBBB the next day after music exposure. Note, that OBBB was observed in all mice (15 of 15) after music exposure (2 h) with intensity 100 dB and in 73% (11 of 15) when music was 90 dB (electronic supplementary material, table S1). Because, our interest was to study the effects of very loud music used in nightclubs, discotheques or rock concerts, where the noise levels are recorded to be over 100 dB [14] (https://www.hear-it.org/disco-noise-volume-over-the-top-1), the next series of experiments were done using the music intensity 100 dB. This choice also was owing to the high reproducibility of music (100 dB)-induced OBBB in all mice.

Despite a quantitative analysis of EBAC is still the most commonly used marker of the BBB integrity, some problems arise from its *ex vivo* and *in vivo* application: about 15% of Evans blue dye remains in the endothelium of cerebral vessels, and Evans blue dye can be taken up by red blood cells and astrocytes [22]. Therefore, for qualitative and quantitative assessment of the BBB integrity we used additional methods to analyse the BBB permeability to low and high molecular weight molecules.

In the second step, further optical measurement experiments employed fluorescein isothiocyanate (FITC)-dextran 70 kDa (hereafter FITCD) injected intravenously 1 h after music exposure (the time of OBBB). The BBB disruption was determined both *ex vivo* using confocal microscopy on brain slices and *in vivo* using two-photon laser scanning microscopy (2PLSM) (figure 1c–e). *Ex vivo* confocal microscopy revealed the accumulation of FITCD outside of brain capillaries 1 h after music exposure. The leakage of FITCD was visualized by fluorescence outside the vessel walls (figure 1d and electronic supplementary material, video S1). No leakage of FITCD was observed in the control group (no music) (figure 1c and electronic supplementary material, video S2). The leakage of FITCD in 2PLSM was also observed 1 h after music exposure and was quantified by measuring the percentage fluorescence intensity of FITCD in the perivascular area (figure 1e). There were no changes in the BBB permeability to FITCD 4 h and 24 h after music exposure.

Figure 1f,g shows the post music BBB disruption measured by magnetic resonance imaging (MRI) of gadolinium-diethylene-triamine-pentaacetic acid (Gd-DTPA) leakage. An analysis using rapid T1 weighting for Gd-DTPA is presented in figure 1g. The image corresponding to 1 h after sound is significantly brighter uniformly than others, indicating leakage of the tracer molecule. This has been quantified by the rate of changes in the MRI signal intensity related to the BBB permeability in consequence from the 1st to 15th scanned images (Ki map) in figure 1f, made for times 1, 4, and 24 h post music stimulus. The data given in figure 1g indicate a statistically significant ($p < 0.001$) increase in the BBB permeability to Gd-DTPA in all regions of the brain 1 h after music exposure. Note that the Ki map values reached 140 ±

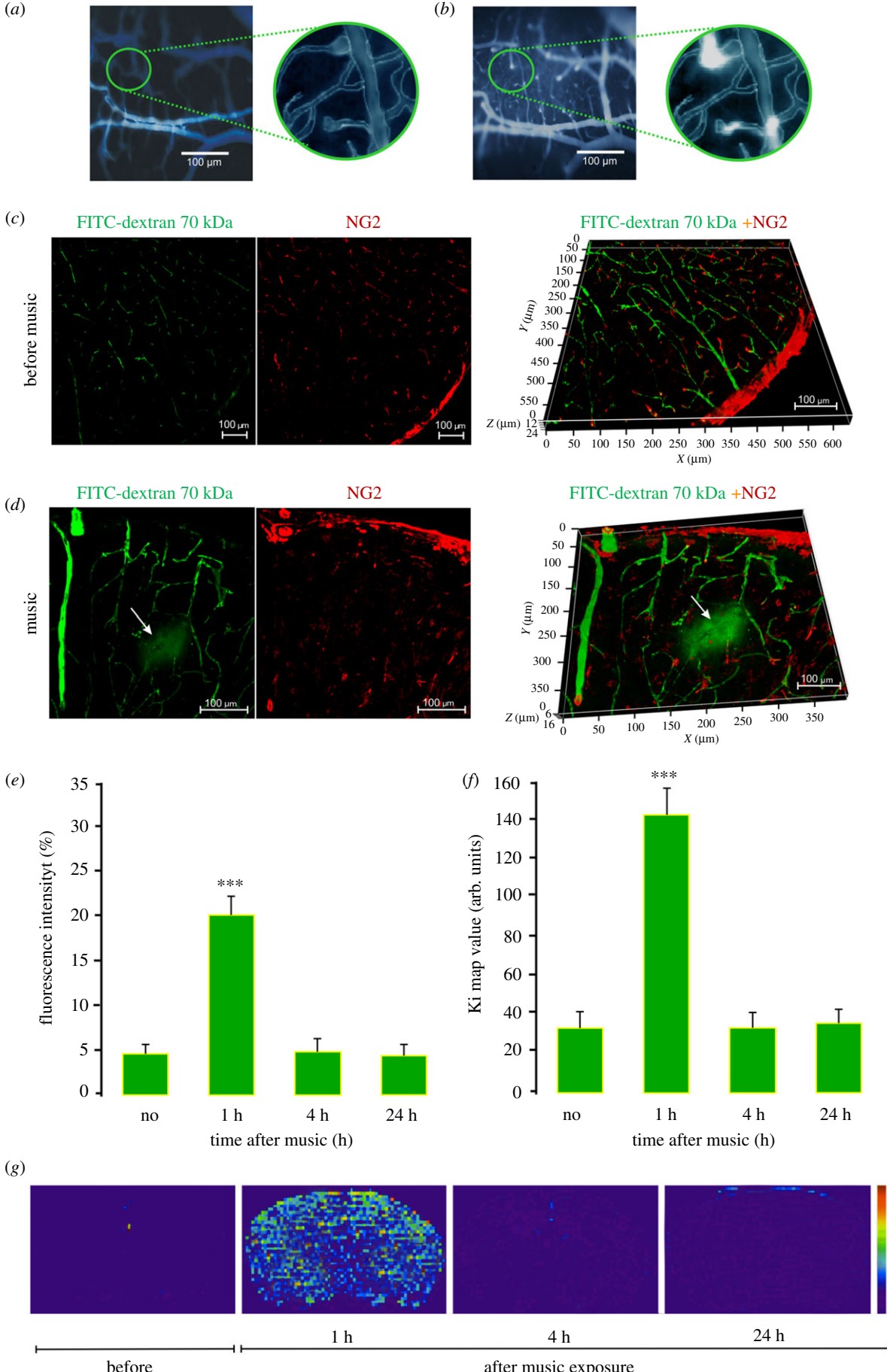

**Figure 1.** (*Caption overleaf.*)

3.7 arbitrary units ($p < 0.001$) only 1 h after music exposure, while after 4 h and 24 h, changes in Ki values were not observed, compared with the control group (no music).

Altogether, the results of our *in vivo* and *ex vivo* experiments clearly demonstrate that loud music exposure (90–100 dB, 11–10 000 Hz, Scorpions 'Still Loving You') during 2 h in

**Figure 1.** (*Overleaf*.) The *ex vivo* and *in vivo* results of music-OBBB (100 dB, 11–10 000 Hz, 2 h intermittent mode: 60 s music; 60 s pause): (*a,b*) *in vivo* real-time fluorescent microscopy of the cerebral microvessels filled with Evans Blue (no EB leakage) before music exposure (*a*) and the EB extravasation from the cerebral capillaries into the brain tissues 1 h after music impact, indicating OBBB (*b*), $n = 15$ in each group; (*c,d*) confocal imaging of brain slices demonstrating the BBB permeability to FITCD in mice subjected to loud music, where (*c*) FITCD intravenous injection but no music exposure (FITCD is constrained to vessels) and (*d*) FITCD injection 1 h after music exposure, substantial leakage indicated by diffuse cloud around vessels, $n = 10$ in each group; (*e*) *in vivo* 2PLSM of the BBB permeability to FITCD in mice subjected to loud music expressed as the percentage fluorescence intensity in the perivascular area. Data are presented as mean ± s.e.m., $n = 10$, ***$p < 0.01$; (*f* and *g*) the MRI analysis of the BBB permeability to Gd-DTPA in mice subjected to loud music, where (*f*) the Ki values show (arbitrary units) rate of changes in the MRI signal intensity, data are presented as mean ± s.e.m., $n = 10$, ***$p < 0.01$, and (*g*) the Ki-maps from one mouse at different time points. (Online version in colour.)

intermittent adaptive mode (1 min – sound; 1 min – pause) is accompanied by delayed (1 h after music impact) and short-lasting (during 1–4 h) OBBB to low and high molecular weight compounds.

## (b) Mechanisms underlying music-induced opening of the blood-brain barrier

For both humans and mice, music significantly affects emotions and behaviour [1,2,23]. There is evidence that mice heard music, which acts as anti-anxiety therapy [23]. However, loud sound triggers a stress response in human and animals [5–8,11–18]. Therefore, we answered the question, what is the role of stress in loud music-induced OBBB. The loud sound caused an increase in the plasma level of the important stress hormone—epinephrine up to 3.1-fold (immediately after music) versus the normal state ($9.0 ± 1.5$ ng ml$^{-1}$ versus $2.9 ± 0.7$ ng ml$^{-1}$, $p < 0.001$, $n = 10$ in each group). One hour after music exposure (the time of OBBB), the level of epinephrine returned almost to the normal value of $3.9 ± 1.6$ ng ml$^{-1}$ ($n = 10$) and was close to the control units 4 h after music impact (the time of BBB closing) ($2.5 ± 0.1$ ng ml$^{-1}$, $n = 10$). These data demonstrate that loud music is the stress factor inducing a transient increase in the serum epinephrine level. However, the BBB was opened 1 h after sound-off, i.e. in the post-stress period when the epinephrine level was restored.

The rise of epinephrine induced by stress is known to increase cerebral blood flow (CBF). This can be the initial factor triggering the BBB leakage [24–27]. To test the cerebrovascular changes associated with music-induced OBBB, we measured relative CBF (rCBF) at the venous (the sagittal sinus) and microcirculatory levels in the same 10 mice before, immediately, 1 h, 4 h and 24 h after sound exposure. Our results demonstrate that immediately after music, rCBF was increased in both venous and microcirculatory levels compared with the control group ($0.80 ± 0.03$ a.u. versus $0.58 ± 0.01$ a.u., $p < 0.001$ for the cerebral microvessels; $1.22 ± 0.01$ a.u. versus $0.83 ± 0.02$ a.u., $p < 0.001$ for the sagittal sinus). One hour after sound exposure, when the BBB was opened, rCBF tended to decrease but continued to be higher than the normal level of rCBF ($0.77 ± 0.08$ a.u. versus $0.58 ± 0.01$ a.u., $p < 0.05$ for the cerebral microvessels; $0.92 ± 0.07$ a.u. versus $0.83 ± 0.02$ a.u., $p < 0.05$ for the sagittal sinus). Four hours after music exposure, when the BBB recovered, rCBF was wtih Evans Blue the normal units ($0.60 ± 0.07$ a.u and $0.58 ± 0.01$ a.u. for the cerebral microvessels; $0.80 ± 0.06$ a.u. and $0.83 ± 0.02$ a.u. for the sagittal sinus, respectively). On the next day, rCBF remained at a normal level ($0.56 ± 0.02$ a.u and $0.58 ± 0.01$ a.u. for the cerebral microvessels; $0.85 ± 0.04$ a.u. and $0.83 ± 0.02$ a.u. for the sagittal sinus, respectively).

The tight junction (TJ) proteins such as claudin-5 (CLDN-5), occluding (OCC) and zonula occludens (ZO-1) play a crucial role in regulation of the BBB permeability [19]. Therefore, we studied the integrity of the TJ proteins immediately after music effects, during OBBB (1 h after music), and in the time of BBB recovery (4 h and 24 h after music) compared with the control group (no music). The data in figure 2a show that the signal intensity from CLDN-5 and OCC but not from ZO-1 were significantly decreased at the time of OBBB. There were no changes in the TJ assembly immediately after listening to music as well as in the time of BBB restoration (4 h and 24 after music).

To study the role of the auditory system in OBBB, we studied the effects of loud music on BBB in deaf mice. The results clearly show no sound effects on the BBB permeability to EBAC in mice with hearing loss, suggesting the important role of the auditory system in a music-induced OBBB ($0.12 ± 0.03$ µg g$^{-1}$ versus $0.10 ± 0.08$ µg g$^{-1}$, respectively, $n = 10$ in each group)).

To analyse the short-term and delayed effects of loud music on the brain and auditory system injury, whole brain and the cochlear, histologic examination was performed using haematoxylin and eosin staining and TUNEL staining for apoptosis 1 h (the time of OBBB), 4 h (the time of BBB closing) and four weeks (delayed effects) after music exposure. The timing of the experiments was dictated by the fact that sound damage may not appear immediately, but after a certain time. For example, peripheral synaptic connections of cochlear neurons are the most vulnerable elements in the cochlea and, in the vast majority of cases, cochlear nerve fibres degenerate only a long time after the sound trauma [28]. In animal models exposed to sound stress, hair cell loss can be widespread within hours [28–32], whereas the loss of the spiral ganglion cells (SGCs) is typically not detectable for several weeks to a month after sound exposure [28,31,32]. The apoptosis evaluation after ultrasound-OBBB is usually detecetd for several hours to four to five weeks after OBBB [33,34]. Therefore, the time of analysis of SGCs and apoptosis was chosen 1 h (the time of OBBB) and four weeks (delayed effects) after music exposure. The histological analysis was performed in the control group (no music), 1 h, 4 h and four weeks after music exposure. The 4 h after music exposure was selected to exclude perivascular edema, which develops during the first hours after OBBB [35].

Figure 2b–e represents cochlear histology and quantification of SGCs density in intact mice (no music), 1 h and four weeks after music exposure. Our results did not reveal any morphological changes in SGCs after short and delayed effects of loud music on the cochlear system. We found that loud music did not induce apoptosis in the mouse brain tissues (data not presented). The histological analysis of brain tissues and the cerebral vessels also did not show any sound-mediated injuries 1 h, 4 h and four weeks after music exposure (electronic supplementary material, figure S5).

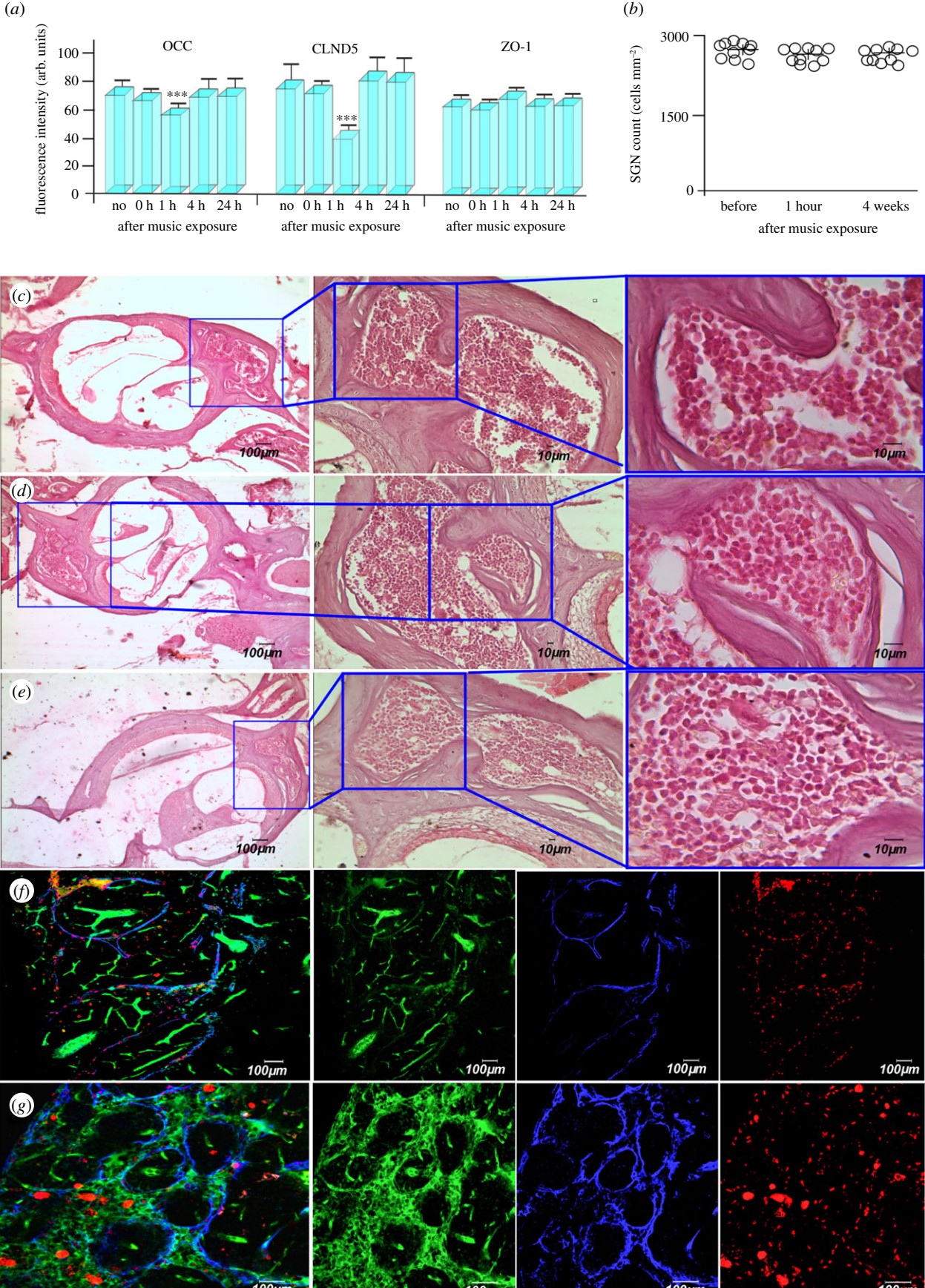

**Figure 2.** Mechanisms underlying music-induced OBBB: (*a*) the signal intensity from TJ proteins in the control group (before music exposure), immediately and 1, 4, 24 h after music impact (*n* = 10 for each group): ***\*\*\*p* < 0.001 versus the control group; (*b*) quantification of SGC cell density before (no music), 1 h and four weeks after music exposure; (*c–e*) cochlear histology: views (64.6×, 246.6×, 774×) of haematoxylin and eosin-stained cochleae of mice before (*c*), 1 h (*d*) and four weeks (*e*) after music impact; (*f,g*) confocal imaging of dcLNs in intact mice (*f*, no music) and in mice with music-OBBB (*g*, 1 h after music exposure): music-OBBB for FITCD was accompanied by lymphatic clearance of FITCD from the brain with its accumulation in the enlarged lymphatic vessels of dcLNs that was not observed in intact mice. (Online version in colour.)

In our previous experiments, we demonstrated that OBBB by photodynamic [35] or infrasound [36,37] effects induces the activation of lymphatic clearance of tracers crossing OBBB that is an important mechanism of brain recovery [38]. Using confocal imaging of the deep cervical lymph nodes (dcLNs), which are the first anatomical station of the cerebral spinal fluid outflow, we clearly demonstrated that music-induced OBBB for FITCD was accompanied by FITCD lymphatic clearance from the brain with its accumulation in the enlarged lymphatic vessels of dcLNs ($22.3 \pm 1.5$ µm versus $37.3 \pm 2.0$ µm, $p < 0.001$) that was not observed in intact mice (figure 2f,g and electronic supplementary material, video S3 and S4).

To exclude effects of anesthesia (2% isoflurane), which we used in *in vivo* experiments (MRI and 2PLSM), on the BBB permeability, we studied the level of EBAC in the brain tissues in mice 30 min after anesthesia (2% isoflurane). We did not find any difference in the EBAC level in the brain between intact and anaesthetized mice ($0.17 \pm 0.01$ µg g$^{-1}$ versus $0.15 \pm 0.01$ µg g$^{-1}$, respectively). This fact allows us to conclude that short time (30 min) and low dose (2% isoflurane) did not affect the BBB permeability in mice.

## 4. Discussion

Music plays a more important role in our life than just being a source of entertainment. For example, music is a powerful therapy that calms down humans and animals [1,2,23]. However, the unsafe listening to loud music triggers the development of NIHR in millions of people, particularly, in teenagers and professional musicians [5,8,11,13–18]. Therefore, it is important to understand how loud music affects us, especially in nightclubs, discotheques and rock concerts, where the continuous sound levels are in excess of 100 dB and are produced for several hours [14] (https://www.hear-it.org/disco-noise-volume-over-the-top-1).

In this experimental study on healthy mice, to our knowledge, for the first time we here demonstrated that the listening of loud rock music (90–100 dB, 11–10 000 Hz, Scorpions 'Still Loving You') is accompanied by delayed (1 h after music exposure) and short-lasting (during 1–4 h) OBBB. Music-induced OBBB for high molecular weight molecules such as EBAC 68.5 kDa and FITCD 70 kDa has been clearly shown by *ex vivo* and *in vivo* experiments using conventional fluorescence microscopy and 2PLSM. Moreover, using MRI, have found OBBB for low molecular weight molecules, Gd-DTPA (928 Da). Note, that mice have hearing with a frequency sensitivity 1–90 kHz, i.e. music used in our experiments is audible for them [39].

We have chosen the intermittent music treatment (1 min sound; 1 min pause) because the safe listening time for a continuous sound of 100 dB is 15 min (https://www.hear-it.org/disco-noise-volume-over-the-top-1). However, there has been no effect found on BBB when mice listened to continuous music of 100 dB during 15 min (electronic supplementary material, table S1). Therefore, the longer music exposure during 1 h and 2 h in a repetitive mode has been chosen for an adaptation to loud sound. When stimuli are repeated, neural activity is usually reduced owing to adaptation [40]. These neural repetition effects have been reported at multiple spatial scales from the level of individual cortical neurons to the level of hemodynamic changes as adaptive mechanisms of the nervous system [41–43]. We have uncovered that a stimulus

period of 2 h at 90–100 dB produced significant OBBB. No changes in the BBB permeability have been found when the stimulus was 70 dB or when 1 h music exposure was used.

Using a 2 h intermittent music exposure, we have found no appearance of apoptotic cells, morphological alterations in the brain tissues, or in SGCs 1 h after music impact when the BBB was opened and 4 h (histological data) after music influences when the BBB integrity was fully restored, or 4 weeks after delayed music effects. We hypothesize that the listening to loud music in the repetitive mode can adapt the brain to sound stress and protect its structure from sound-induced injuries.

The frequencies range of music (Scorpions 'Still Loving You') was 11–10 000 Hz and maximal intensity was around 100 Hz (electronic supplementary material, figure S2). The mice's hearing sensitivity is between 1 K to 90–100 kHz [39]. Thus, the maximum intensity frequency of 100 Hz for mice is infrasound (IS) for mice. The IS-induced opening of BBB has been shown in other experimental works [44,45]. However, there are no findings demonstrating direct IS effect on the BBB permeability in humans. There is a growing evidence that humans are indeed receptive to IS and that exposure to low-frequency sounds can give rise to high levels of annoyance, distress, sleep disturbances, headache and dizziness, from tinnitus and hyperacusis, to panic attacks and depression [46]. However, Leventhall [47, p. 135] concluded that 'if you cannot hear a sound and you cannot perceive it in other ways and it does not affect you'. The World Health Organization suggests 'there is no reliable evidence that ISs below the hearing threshold produce physiological or psychological effects' [48, p. 4]. It appears that the notion, according to which sound needs to be perceived in order to exert relevant effects on the organism, falls short when aiming at an objective risk assessment of IS, especially if one takes into consideration recent advances in research on inner ear physiology as well as on the effects of subliminal auditory stimulation (i.e. stimulation below the threshold of perception). So, 5-Hz IS exposure (60–65 dB) has been shown to trigger the response of inner ear components such as the outer hair cells in animals and it has been suggested that outer hair cell stimulation may also exert a broader influence on the nervous system via the brainstem [49]. In addition, there is the well documented effect in cognitive science that brain physiology and behaviour can be influenced by a wide range of subliminally presented stimuli, including stimuli of the auditory domain [50]. The IS near the hearing threshold induces changes of neural activity across several brain regions, some of which are known to be involved in auditory processing, while others are regarded as keyplayers in emotional and vascular autonomic control in humans [51].

We have presented in detail possible systemic and molecular mechanisms responsible for music-induced OBBB. It is known that stress hormones, such as epinephrine, might induce an increase in the BBB permeability via vasodilation of cerebral vessels and an increase in the extension of cracks in the tight contacts of endothelial cells with a change in the ultrastructure of the astrocyte end-feet; increase transport and pinocytotic activity of endothelial cells [24–27]. The elevated stress hormone levels induced by sound were demonstrated in humans and animals in [52]. Our results indicate that loud music induces a rise of serum epinephrine with a significant increase in CBF at both macro- and microcirculatory levels. We assume that the elevation of epinephrine by stress causes an increase of CBF and changes in the tone of cerebral vessels that could be an initial factor triggering the BBB leakage.

Indeed, 1 h after music exposure, we have observed a decrease of the signal intensity from the TJ proteins, such as CLND5 and OCC, suggesting a temporal disorganization of the TJ assembly. It is important to note that already 4 h after music effects, the signal intensity from the TJ proteins was within normal units that were preserved the next day after the experiment. We hypothesize that these fast changes can be explained by the internalization of the TJ proteins resulting in a temporal loss of their surface in the space between endothelial cells that can be one of the mechanisms underlying music-induced OBBB. Our hypothesis is very close to the mechanism of BBB disruption via the internalization of vascular endothelial cadherin cadherin induced by vascular endothelial growth factor [53]. However, our findings cannot demonstrate the precise time of BBB recovery owing to the high variability of BBB closing in different mice. It appears that the process of BBB recovery can already start during stress owing to corticosterone-mediated reduction of BBB permeability [54].

Using a model of sensorineural deafness in mice, we have not found an effect of sound on the BBB permeability to EBAC in mice with a hearing loss, suggesting an important role of the auditory system in music-induced OBBB.

In summary, our findings clearly demonstrate that loud music induces a temporal OBBB with a fast recovery of BBB functions. The OBBB is accompanied by lymphatic clearance of molecules penetrating into the brain via OBBB that can be a crucial mechanism of quick restoration of the BBB integrity after music exposure [35–37,55].

## 5. Conclusion

This pioneering experimental study has discovered the important phenomenon of OBBB in healthy mouse brain induced by loud rock music (90–100 dB, 11–10 000 Hz, Scorpions 'Still Loving You'). The listening of loud music during 2 h in an intermittent adaptive mode is accompanied by delayed (1 h after music exposure) and short-lasting (during 1–4 h) OBBB for high and low molecular weight molecules. We assume that an elevation of epinephrine by sound stress causes an increase of CBF and changes in the tone of cerebral vessels that could be an initial factor triggering the BBB leakage associated with a decrease in expression of TJ proteins. The music-induced OBBB is reversible with quick restoration of the BBB function without the brain tissue and the cochlea injuries. It can be related to the repetitive mode of music exposure and post-OBBB activation of lymphatic clearance of molecules penetrating into the brain via OBBB.

Our results stimulate a revision of the traditional knowledge about the BBB nature. The BBB opens itself in a healthy brain as a response to sound stress. Despite the fact that we did not find any injuries of the brain tissues and the cochlea after loud music-OBBB owing to limitation of our studies of delayed music effects by four weeks, even temporal OBBB can be harmful for the brain via opening the door for viruses, bacteria and toxins. It is important to note that millions of teenagers and professional rock, jazz and symphony orchestra musicians are in a potential risk group of OBBB owing to exposure to loud music in concerts, nightclubs, discotheques and pubs. Therefore, more detailed studies of loud music-induced OBBB are required for further investigations in humans.

We also believe that our findings can be the basis for novel strategies in optimization of sound-mediated methods for brain drug delivery. Despite the fact that music opens BBB in a non-targeted manner, loud music has a high potential for clinical applications as an easily used, non-invasive, low cost, labelling free, perspective and completely new approach for the treatment of neurodegenerative disorders, such as Alzheimer's disease or myotrophic lateral sclerosis, which are associated with injuries of many brain regions and diffusive progression without specificity of the brain areas. We address further research of music-induced OBBB as a proof of concept for exploration of sound-induced drug delivery to the brain. Such issues as the study of efficacy of the drug delivery into the brain and the effect of different levels of sound on the pharmacokinetics of the delivered drug can shed light on the development of a new alternative method of drug brain delivery for different neurodegenerative diseases, brain oncology and brain trauma.

This pilot study has limitations including using only one music composition, such as Scorpions' 'Still Loving You', which we tested in varying loudness intensities (70 dB, 90 dB and 100 dB), durations (15 min, 30 min, 1 h and 2 h after music exposure), deliveries (60 s music; 60 s pause) and possible systemic and molecular mechanisms underlying phenomenon of loud music-induced OBBB. In further experiments, we will study the effects of other musical stimulus on BBB, focusing on the analysis of effects of musical elements in comparison of timbre, harmony, rhythm, tempo, texture, homophony, heterophony using instrumental/vocal combinations for better understanding of complex music effects on the BBB permeability.

Ethics. The experiments were conducted on male mongrel mice (20–25 mg). All procedures were performed in accordance with the Guide for the Care and Use of Laboratory Animals. The experimental protocols were approvied by the local Bioetics Commission of the Saratov State University (protocol no. 7) and the Institutional Animal Care and Use Committee of the University of New Mexico, USA (200247).

Data accessibility. Data accessibility section including MRI, CBF, EBAC, apoptosis, histological, inner ear, confocal data of OBBB, confocal data of the dcLN, ARB threshold SPL, fluorescent imaging of EBAC, and signal intensity from TJ data are available from the Dryad Digital Repository: https://doi.org/10.5061/dryad.wwpzgmshr [56].

Competing interests. We declare we have no competing interests.

Funding. S.-G.O., A.S., N.N. and J.K. were supported by RF Governmental grant no. 075–15-2019-1885, grant from RSF no. 20-15-00090 and 19-15- 00201, grant from RFBR 19-515-55016 China a, 20-015-00308-a. D.B. was supported by NIH R01 NS112808 and R21NS091600.

Acknowledgments. We would like to express our special thanks of gratitude to Prof. John Connor (University of New Mexico) for discussion of our results and much helpful advices for preparation of the manuscript.

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
