## [Reviewer comments · Proceedings of the Royal Society B: Biological Sciences]

Review History

RSPB-2020-0648.R0 (Original submission)

Review form: Reviewer 1

Recommendation

Major revision is needed (please make suggestions in comments)

Scientific importance: Is the manuscript an original and important contribution to its field?

Good

General interest: Is the paper of sufficient general interest?

Good

Quality of the paper: Is the overall quality of the paper suitable?

Acceptable

Is the length of the paper justified?

Yes

Should the paper be seen by a specialist statistical reviewer?

No

Do you have any concerns about statistical analyses in this paper? If so, please specify them explicitly in your report.

No

It is a condition of publication that authors make their supporting data, code and materials available - either as supplementary material or hosted in an external repository. Please rate, if applicable, the supporting data on the following criteria.

Is it accessible?

No

Is it clear?

No

Is it adequate?

No

Do you have any ethical concerns with this paper?

No

Comments to the Author

Please see attached file. (See Appendix A)

Review form: Reviewer 2

Recommendation

Major revision is needed (please make suggestions in comments)

Scientific importance: Is the manuscript an original and important contribution to its field?

Good

General interest: Is the paper of sufficient general interest?

Good

Quality of the paper: Is the overall quality of the paper suitable?

Acceptable

Is the length of the paper justified?

Yes

Should the paper be seen by a specialist statistical reviewer?

No

Do you have any concerns about statistical analyses in this paper? If so, please specify them explicitly in your report.

Yes

It is a condition of publication that authors make their supporting data, code and materials available - either as supplementary material or hosted in an external repository. Please rate, if applicable, the supporting data on the following criteria.

Is it accessible?

No

Is it clear?

N/A

Is it adequate?

N/A

Do you have any ethical concerns with this paper?

No

Comments to the Author

The authors investigated the effect of music/sound on the blood-brain barrier using a mouse model. They found that two hours of 90-100 dB music/sound safely open the BBB to low/high molecular weight molecules in mice. The authors argue that BBB opens after one hour, and closes after 4 or 24 hours. The results of this study could have a potential impact on non-invasive therapy for drug delivery in humans. The authors used a rigorous analysis to address the mechanism of BBB opening as a result of sound exposure.

I have comments and suggestions aimed to address some conceptual problems derived from the experimental approach and also sought to improve the presentation of the paper. In this study, the authors argue that both music and sound open BBB. Except for results regarding permeability to the Evans Blue Albumin Complex, all data in the figures are shown for the sound.

Music of 90-100 dB (11-10,000 Hz) and sound of 90-100 dB (370Hz) are different stimuli since the mice hearing sensitivity is between 1K-100kHz vs. human's one (100Hz-20kHz). Thus, the sound (370Hz) used in this study is infrasound for mice. What was the reason for authors to chose the infrasound as a stimulus in the experiments? What changes this sound produce in cochlea? The fact that authors used infrasound in this study and observed the opening of BBB could suggest a different mechanism of opening of BBB, for example, via mechanoreceptors. There are studies showing that infrasound affects the outer hair cells but not inner hair cells. This should be addressed in the discussion. The authors should present data for music as well and compare those to sound one.

The authors argue that "the BBB disruption caused by sound is associated with a post-stress period (1h after sound exposure) when the serum epinephrine level was significantly decreased compared with stress". (please, correct typo in "epinephrine", "decreased"). However, they tested BBB in sound experiments after one hour, not immediately after. The rise in epinephrine induced by stress is known to increase cerebral blood flow and to cause leaks in the BBB. Is it possible that the BBB opening was initiated earlier and could be explained, at least partially, by the rise of epinephrine? In support, their data showed that immediately after the sound, "the CBF was increased in both venous and microcirculatory levels" (fig 2b).

The authors argue that this approach is safe and did not produce hearing problems in mice. They used the fear conditioning test to assess the hearing status. Many studies are suggesting that sound above 85dB could be damaging for the hair cells. The fear conditioning test does not reflect changes in hair cells of the cochlea. Also, the authors argue that this could be used in humans. Two hours of 90-100 dB music/sound exposure is quite a long time, and potentially, it could be harmful. Authors should be cautious when they discuss it.

The manuscript contains multiply typos and errors that need to be corrected.

Page 2. In Author-supplied statements. Data. To question, "It is a condition of publication that data, code, and materials supporting your paper are made publicly available. Does your paper present new data?" The authors wrote, "My paper has no data." Please, explain.

Page 7. Introduction. After "focused ultrasound," add abbreviation because later authors refer to FUS, and it could be confusing.

Page 7. Introduction. Authors wrote, "There are no special centers, which are involved in the development of AD or ASL. Therefore, the brain region for therapy of these diseases and targeted FUS-related brain drug delivery is not clear." It is a confusing statement because many research centers are focusing on studying Alzheimer's disease and brain regions that are involved in developing this disease. The authors need to elaborate on their statement.

The word "involved" misspelled.

Page 14. Methods. "Evans Blue dye (Sigma Chemical Co., St. Louis, Michigan". Sigma Chemical Co is located in St. Louis, Missouri. Please, correct it.

Page 16. Methods. In Assessment of apoptosis with TUNEL method. "The number of apoptotic cells before and in 1h after sound exposure (n=7 in each group) was evaluated..". Please, explain why this analysis was conducted only in 1h after sound exposure.

Page 8. Results. Authors wrote, "Thus, this series of experiments demonstrated that loud music/sound (100 dB) reversibly opens the BBB to EBAC in all tested mice and these changes are similar either for music or sound". This conclusion is confusing since authors found that only "stimulus period of 2 h at 90-100 dB produced a robust increase in the BBB", while 70 dB was ineffective and "4h and 24h after either type of stimulation the BBB permeability to EBAC had returned to levels near the normal state..". The authors should be more specific and accurate in the description of the results.

Page 8. Results. "Figures 1, o,n show post sound the BBB disruption.." There is no Figure 1 o.

Page. 10. In Results. "In the first step, we answered the question, what is the role of stress in sound-induced BBB opening. The loud sound caused an increase in the plasma level of important stress hormone such as epinephrine up to 7.0-fold (immediately after sound) vs. the normal state (23.1±2.7 ng/ml vs. 3.3±0.9 ng/ml, p<0.001, n=10 in each group)". The authors need to clarify to the time of the measurement and how long the music/sound was delivered (0.25, 1 h or 2h).

Page. 10. results. The "audible system" needs to be correct to the "auditory system" throughout the text.

Page 13. Discussion, "acostically", correct the typo.

Figure 2 has a poor legend description, lack of abbreviation, errors. "e - The schema of mechanisms of sound-induced BBB opening". It should be "d" and correct to "schematic illustration of...". Add all necessary abbreviations, and it is not clear to what "before" and "After" sound is related to what figure? In Figure 2b, define blue and red color.

Page 11. The authors wrote, "Our data are consistent with the other results suggesting an increase in CBF via dilation of the cerebral vessels in humans and in the cochlear blood flow in the guinea pig after the electrical stimulation of the vestibular nerve [13,14]". This is a very confusing and misleading conclusion since this manuscript does not study the vestibular activation at all. The study [ref. 13] addresses precisely how the stimulation of the vestibular system by tilting and translating subject changes the cerebral blood flow. The authors in ref 14 studied cochlear blood flow in the guinea pig by stimulating different parts of the cochlea and cervical ganglion.

Decision letter (RSPB-2020-0648.R0)

29-Jun-2020

Dear Dr Semachkina-Glushkovskaya:

I am writing to inform you that your manuscript RSPB-2020-0648 entitled "Music/sound opens the blood-brain barrier: a readily available approach to brain drug delivery and therapy of brain diseases" has, in its current form, been rejected for publication in Proceedings B.

This action has been taken on the advice of referees, who have recommended that substantial revisions are necessary. With this in mind we would be happy to consider a resubmission,

provided the comments of the referees are fully addressed. However please note that this is not a provisional acceptance.

Sincerely,
Dr Robert Barton
<mailto:proceedingsb@royalsociety.org>

Associate Editor
Board Member: 1
Comments to Author:

Two expert reviewers have now seen your manuscript, and while both are positive regarding your new results, they each have significant concerns about how your data have been presented and interpreted and how likely the raw data will become available once the manuscript is published. In particular, both reviewers are concerned about what affects the levels of sound (and indeed the frequency of sound) might have on the health of the cochlea and thus on the likelihood of your new method being used on humans. They were also concerned about the interpretation of the data and how the data have been presented and discussed. All these issues need to be addressed prior to publication.

Reviewer(s)' Comments to Author:
Referee: 1

Comments to the Author(s)
Please see attached file.

Referee: 2

Comments to the Author(s)
The authors investigated the effect of music/sound on the blood-brain barrier using a mouse model. They found that two hours of 90-100 dB music/sound safely open the BBB to low/high molecular weight molecules in mice. The authors argue that BBB opens after one hour, and closes after 4 or 24 hours. The results of this study could have a potential impact on non-invasive

therapy for drug delivery in humans. The authors used a rigorous analysis to address the mechanism of BBB opening as a result of sound exposure.

I have comments and suggestions aimed to address some conceptual problems derived from the experimental approach and also sought to improve the presentation of the paper. In this study, the authors argue that both music and sound open BBB. Except for results regarding permeability to the Evans Blue Albumin Complex, all data in the figures are shown for the sound.

Music of 90-100 dB (11-10,000 Hz) and sound of 90-100 dB (370Hz) are different stimuli since the mice hearing sensitivity is between 1K-100kHz vs. human's one (100Hz-20kHz). Thus, the sound (370Hz) used in this study is infrasound for mice. What was the reason for authors to chose the infrasound as a stimulus in the experiments? What changes this sound produce in cochlea? The fact that authors used infrasound in this study and observed the opening of BBB could suggest a different mechanism of opening of BBB, for example, via mechanoreceptors. There are studies showing that infrasound affects the outer hair cells but not inner hair cells. This should be addressed in the discussion. The authors should present data for music as well and compare those to sound one.

The authors argue that "the BBB disruption caused by sound is associated with a post-stress period (1h after sound exposure) when the serum epinephrine level was significantly decreased compared with stress". (please, correct typo in "epinephrine", "decreased"). However, they tested BBB in sound experiments after one hour, not immediately after. The rise in epinephrine induced by stress is known to increase cerebral blood flow and to cause leaks in the BBB. Is it possible that the BBB opening was initiated earlier and could be explained, at least partially, by the rise of epinephrine? In support, their data showed that immediately after the sound, "the CBF was increased in both venous and microcirculatory levels" (fig 2b).

The authors argue that this approach is safe and did not produce hearing problems in mice. They used the fear conditioning test to assess the hearing status. Many studies are suggesting that sound above 85dB could be damaging for the hair cells. The fear conditioning test does not reflect changes in hair cells of the cochlea. Also, the authors argue that this could be used in humans. Two hours of 90-100 dB music/sound exposure is quite a long time, and potentially, it could be harmful. Authors should be cautious when they discuss it.

The manuscript contains multiply typos and errors that need to be corrected.

Page 2. In Author-supplied statements. Data. To question, "It is a condition of publication that data, code, and materials supporting your paper are made publicly available. Does your paper present new data?" The authors wrote, "My paper has no data." Please, explain.

Page. 7. Introduction. After "focused ultrasound," add abbreviation because later authors refer to FUS, and it could be confusing.

Page 7. Introduction. Authors wrote, "There are no special centers, which are involved in the development of AD or ASL. Therefore, the brain region for therapy of these diseases and targeted FUS-related brain drug delivery is not clear." It is a confusing statement because many research centers are focusing on studying Alzheimer's disease and brain regions that are involved in developing this disease. The authors need to elaborate on their statement. The word "involved" misspelled.

Page 14. Methods. "Evans Blue dye (Sigma Chemical Co., St. Louis, Michigan". Sigma Chemical Co is located in St. Louis, Missouri. Please, correct it.

Page 16. Methods. In Assessment of apoptosis with TUNEL method. "The number of apoptotic cells before and in 1h after sound exposure (n=7 in each group) was evaluated..". Please, explain why this analysis was conducted only in 1h after sound exposure.

Page 8. Results. Authors wrote, "Thus, this series of experiments demonstrated that loud music/sound (100 dB) reversibly opens the BBB to EBAC in all tested mice and these changes are similar either for music or sound". This conclusion is confusing since authors found that only "stimulus period of 2 h at 90-100 dB produced a robust increase in the BBB", while 70 dB was ineffective and "4h and 24h after either type of stimulation the BBB permeability to EBAC had returned to levels near the normal state..". The authors should be more specific and accurate in the description of the results.

Page 8. Results. "Figures 1, o,n show post sound the BBB disruption.." There is no Figure 1 o.

Page. 10. In Results. "In the first step, we answered the question, what is the role of stress in sound-induced BBB opening. The loud sound caused an increase in the plasma level of important stress hormone such as epinephrine up to 7.0-fold (immediately after sound) vs. the normal state (23.1 ± 2.7 ng/ml vs. 3.3 ± 0.9 ng/ml, $p < 0.001$, $n = 10$ in each group)". The authors need to clarify to the time of the measurement and how long the music/sound was delivered (0.25, 1 h or 2h).

Page. 10. results. The "audible system" needs to be correct to the "auditory system" throughout the text.

Page 13. Discussion, "acoustically", correct the typo.

Figure 2 has a poor legend description, lack of abbreviation, errors. "e - The schema of mechanisms of sound-induced BBB opening". It should be "d" and correct to "schematic illustration of...". Add all necessary abbreviations, and it is not clear to what "before" and "After" sound is related to what figure? In Figure 2b, define blue and red color.

Page 11. The authors wrote, "Our data are consistent with the other results suggesting an increase in CBF via dilation of the cerebral vessels in humans and in the cochlear blood flow in the guinea pig after the electrical stimulation of the vestibular nerve [13,14]". This is a very confusing and misleading conclusion since this manuscript does not study the vestibular activation at all. The study [ref. 13] addresses precisely how the stimulation of the vestibular system by tilting and translating subject changes the cerebral blood flow. The authors in ref 14 studied cochlear blood flow in the guinea pig by stimulating different parts of the cochlea and cervical ganglion.

Author's Response to Decision Letter for (RSPB-2020-0648.R0)

See Appendices B & C.

RSPB-2020-2337.R0

Review form: Reviewer 2

Recommendation

Accept with minor revision (please list in comments)

Scientific importance: Is the manuscript an original and important contribution to its field?

Good

General interest: Is the paper of sufficient general interest?

Excellent

Quality of the paper: Is the overall quality of the paper suitable?

Good

Is the length of the paper justified?

Yes

Should the paper be seen by a specialist statistical reviewer?

No

Do you have any concerns about statistical analyses in this paper? If so, please specify them explicitly in your report.

No

It is a condition of publication that authors make their supporting data, code and materials available - either as supplementary material or hosted in an external repository. Please rate, if applicable, the supporting data on the following criteria.

Is it accessible?

Yes

Is it clear?

Yes

Is it adequate?

Yes

Do you have any ethical concerns with this paper?

No

Comments to the Author

This is a revised version of the manuscript previously titled "Music/sound opens the blood-brain barrier: a readily available approach to brain drug delivery and therapy of brain diseases." The authors changed the presentation leaving only data regarding the effect of music on the BBB. The authors revised the manuscript's primary focus from presenting it as a search for a non-invasive approach for increasing BBB permeability to describing a phenomenon of BBB opening during the music in general. They carefully discussed this phenomenon as a potential usage in drug-delivery therapy at the end. I agree with these changes because more studies are needed to understand the BBB opening mechanisms during loud music. Loud music/sound could potentially negatively affect the hearing. Overall, the author substantially improved the manuscript in all aspects: discussion, methods, modified and added information in figures, and corrected many other issues.

My main comment is about the description of the main findings. Authors found that exposure to music 90-100dB during 2 hours increases the BBB permeability and that significant changes in BBB were observed at 1 hour after music-off time. The authors wrote, "This method also revealed the delayed OBBB that was only 1-4 h after music effects but not immediately music-off". It is confusing because, at 4 hours after the music stopped, the BBB activity did not differ from the control group. Thus, based on the available data, the authors cannot specify how long the BBB stayed opened: 15 min, one hour, or 2-3 hours. For this, more data need to be collected at a different time (1,2,3, hours). If authors have additional data for time between 1-4 hours, then they could present it. If they do not have it, they need to state what they found precisely.

In Fig 2SI, the authors showed "the frequency range of music (Scorpions "Still Loving You"): frequencies in the range of 11-10,000 Hz and maximal intensity around 100 Hz". The mice's hearing sensitivity is between 1K-100kHz. Thus the maximum intensity frequency of 100Hz would still be infrasound. It would be essential to compare to humans because mice experienced this music with less intensity than humans.

When authors discuss the possible role of epinephrine in the BBB opening, it is important to mention the role of corticosterone as another stress hormone. It has been shown that corticosterone has the opposite effect on BBB and tighten BBB. Thus, it could be that both hormones contribute to regulation of BBB during stress, in this case, loud music.

Minor comments:

Add to the manuscript that you used fluorescein isothiocyanate (FITC) to assess permeability to macromolecules.

Inconsistent usage of the abbreviation opening of BBB, sometimes it is OBBB and sometimes BBBO.

In Fig.1 The Ki-maps from the same mice at different time points. Is it superimposed maps from all mice or one example mouse?

Page 9, line 5. "4h (histological data)". I believe it should be for 4 weeks.

Suppl material, table 1. Data for 90dB. n=11, legend said n = 10. Also, it is not a clear # of 4 animals. In 4 mice, BBB did not open after 4 hours or after 1 hour?

In supplemental material, add the description to videos.

Page 1. Correct typo in " All procedures were performed in accordance with the "Guide for the Care and Use of Laboratory Animals". The experimental protocols were approved by the Local Bioethics Commission of the Saratov State University (Protocol No. 7); the Institutional Animal Care and Use Committee of the University of New Mexico, USA (#200247).

Review form: Reviewer 3

Recommendation

Accept with minor revision (please list in comments)

Scientific importance: Is the manuscript an original and important contribution to its field?

Excellent

General interest: Is the paper of sufficient general interest?

Excellent

Quality of the paper: Is the overall quality of the paper suitable?

Excellent

Is the length of the paper justified?

Yes

Should the paper be seen by a specialist statistical reviewer?

No

Do you have any concerns about statistical analyses in this paper? If so, please specify them explicitly in your report.

No

It is a condition of publication that authors make their supporting data, code and materials available - either as supplementary material or hosted in an external repository. Please rate, if applicable, the supporting data on the following criteria.

Is it accessible?

Yes

Is it clear?

Yes

Is it adequate?

Yes

Do you have any ethical concerns with this paper?

No

Comments to the Author

Semyachkina-Glushkovskaya et al. provide a putative mechanistic explanation behind the use of rock music to affect blood-brain barrier permeability (BBB-p) in mice. They used Scorpions' "Still Loving You" at 70 dB, 90 dB, and 100dB at .25hr, 1hr, and 2hr time points, to determine BBB permeability to Evans Blue Albumin Complex (EBAC, 68.5 kDa) between 1-4hrs after exposure. This work is most innovative, and there is practically the only team that is addressing such questions. They have a history of rigorous research.

In and Ex vivo real-time fluorescent microscopy analysis confirmed highest concentrations of EBAC within cerebral microvessels after 1hr exposure to music (100 dB, 11-10,000 Hz, 2h intermittent mode: 60 sec – music; 60 sec – pause). A second analysis of BBB-p involved ex vivo confocal microscopy on brain slices and in vivo two-photon laser scanning microscopy (2PLSM). Ex vivo confocal microscopy revealed the accumulation of FITCD outside of brain capillaries 1h after music exposure. No leakage of FITCD was observed in the control group (no music). The leakage of FITCD in 2PLSM was also observed 1h after music exposure and was quantified by measuring the percentage fluorescence intensity of FITCD in the perivascular area. A third analysis of BBB-p involved magnetic resonance imaging (MRI) of gadolinium-diethylenetriamine-pentaacetic acid (Gd-DTPA) leakage. Results indicate a statistically significant ($p < .001$) increase in the BBB permeability to Gd-DTPA in all regions of the brain 1h after music exposure. Altogether, the results demonstrate that loud music exposure (90-100 dB, 11 – 10,000 Hz, Scorpions "Still loving you") during 2 hrs in intermittent adaptive mode (1 min – sound; 1 min – pause) is accompanied by delayed (1h after music impact) and short-lasting (during 1-4 hrs) OBBB to low and high molecular weight compounds.

The stress response to loud music involved the release of epinephrine, as confirmed by an increase in the plasma level of epinephrine up to 3.1-fold (immediately after the music) vs. the normal state (9.0 ± 1.5 ng/ml vs. 2.9 ± 0.7 ng/ml, $p < .001$, $n=10$ in each group). One hour after music exposure (the time of OBBB), the level of epinephrine returned almost to the normal value of 3.9 ± 1.6 ng/ml ($n=10$) and was over the control units 4h after music impact (the time of BBB closing) (2.5 ± 0.1 ng/ml, $n=10$). To test this, they measured changes in cerebral blood flow at the venous (the Sagittal sinus) and microcirculatory levels in the same 10 mice before, immediately, 1h, 4h, and 24h after sound exposure. The results demonstrate that immediately after music rCBF was increased in both venous and microcirculatory levels compared with the control group (0.80 ± 0.03 a.u. vs. 0.58 ± 0.01 a.u., $p < .001$ for the cerebral microvessels; 1.22 ± 0.01 a.u. vs. 0.83 ± 0.02 a.u., $p < .001$ for the Sagittal sinus). One hour after sound exposure, when the BBB was opened, rCBF tended to decrease but continued to be higher than the normal level of rCBF (0.77 ± 0.08 a.u. vs. 0.58 ± 0.01 a.u., $p < .05$ for the cerebral microvessels; 0.92 ± 0.07 a.u. vs. 0.83 ± 0.02 a.u., $p < .05$ for the Sagittal sinus).

Finally, they studied integrity of tight junction proteins (claudin-5 (CLDN-5), occluding (OCC) and zonula occludens (ZO-1) immediately after music effects, during OBBB (1h after the music), and in the time of BBB recovery (4h and 24h after the music) compared with the control group (no music). Results show that the signal intensity from CLDN-5 and OCC but not from ZO-1 were significantly decreased in the time of OBBB. There were no changes in the TJ assembly immediately after listening to music as well as in the time of BBB restoration (4h and 24 after the music). Importantly, music-OBBB for FITCD was accompanied by lymphatic clearance of FITCD from the brain with its accumulation in the enlarged lymphatic vessels of dcLNs (deep cervical lymph nodes) that was not observed in intact mice. Using confocal imaging of the deep cervical lymph nodes (dcLNs), which are the first anatomical station of the cerebral spinal fluid (CSF) outflow, they show that music-induced OBBB for FITCD was accompanied by FITCD lymphatic clearance from the brain with its accumulation in the enlarged lymphatic vessels of dcLNs (22.3 ± 1.5 μ m vs. 37.3 ± 2.0 μ m, $p < .001$) that was not observed in intact mice.

The limitation section could be expanded. For example, variability in the potential effects of the complexity of the music used in this study may have skewed results. The musical selection used in this study was Scorpions' "Still Loving You." The authors tested varying loudness levels in decibels (70, 90, and 100), varying durations (.25hr, 1hr, and 2hr), and varying deliveries (60sec music - 60sec pause). Comparison/contrast of effect(s) upon BBB-p of other musical stimuli/selections was not conducted. Such a comparison/contrast may have concretely concluded that volume level is indeed the sole reason for BBB-p, rather than effects of other musical elements (e.g., melody, timbre, harmony, rhythm, tempo, texture, form, etc.), and/or combination of volume with other musical elements upon BBB-p. Inclusion of comparator musical piece(s)/song(s) involving distinctly different musical qualities to "Still Loving You" would provide complementary evidence of the efficacy of S-G et al. (2020) use of this song to draw their conclusions. Future work should include a piece(s) of the music of distinctly different tempo/rhythm, key signature/mode, instrumentation/timbre/arrangement, and/or texture (e.g., homophony, heterophony, etc.). Furthermore, instrumental vs. vocal vs. instrumental/vocal combinations could be tested to concretely exclude these variables.

Decision letter (RSPB-2020-2337.R0)

10-Nov-2020

Dear Dr Semachkina-Glushkovskaya

I am pleased to inform you that your Review manuscript RSPB-2020-2337 entitled "Phenomenon of music-induced opening of the blood-brain barrier in healthy mice" has been accepted for publication in Proceedings B.

The referee(s) have recommended publication, but also suggest some minor revisions to your manuscript. Therefore, I invite you to respond to the referee(s)' comments and revise your manuscript. Because the schedule for publication is very tight, it is a condition of publication that you submit the revised version of your manuscript within 7 days. If you do not think you will be able to meet this date please let us know.

- 1) A text file of the manuscript (doc, txt, rtf or tex), including the references, tables (including captions) and figure captions. Please remove any tracked changes from the text before submission. PDF files are not an accepted format for the "Main Document".

2) A separate electronic file of each figure (tiff, EPS or print-quality PDF preferred). The format should be produced directly from original creation package, or original software format. PowerPoint files are not accepted.

3) Electronic supplementary material: this should be contained in a separate file and where possible, all ESM should be combined into a single file. All supplementary materials accompanying an accepted article will be treated as in their final form. They will be published alongside the paper on the journal website and posted on the online figshare repository. Files on figshare will be made available approximately one week before the accompanying article so that the supplementary material can be attributed a unique DOI.

Sincerely,

Professor Innes Cuthill

Reviews Editor, Proceedings B

Reviewer(s)' Comments to Author:

Referee: 2

Comments to the Author(s).

This is a revised version of the manuscript previously titled "Music/sound opens the blood-brain barrier: a readily available approach to brain drug delivery and therapy of brain diseases." The authors changed the presentation leaving only data regarding the effect of music on the BBB. The authors revised the manuscript's primary focus from presenting it as a search for a non-invasive approach for increasing BBB permeability to describing a phenomenon of BBB opening during the music in general. They carefully discussed this phenomenon as a potential usage in drug-delivery therapy at the end. I agree with these changes because more studies are needed to understand the BBB opening mechanisms during loud music. Loud music/sound could potentially negatively affect the hearing. Overall, the author substantially improved the manuscript in all aspects: discussion, methods, modified and added information in figures, and corrected many other issues.

My main comment is about the description of the main findings. Authors found that exposure to music 90-100dB during 2 hours increases the BBB permeability and that significant changes in BBB were observed at 1 hour after music-off time. The authors wrote, "This method also revealed the delayed OBBB that was only 1-4 h after music effects but not immediately music-off". It is confusing because, at 4 hours after the music stopped, the BBB activity did not differ from the control group. Thus, based on the available data, the authors cannot specify how long the BBB stayed opened: 15 min, one hour, or 2-3 hours. For this, more data need to be collected at a different time (1,2,3, hours). If authors have additional data for time between 1-4 hours, then they could present it. If they do not have it, they need to state what they found precisely.

In Fig 2SI, the authors showed "the frequency range of music (Scorpions "Still Loving You"): frequencies in the range of 11-10,000 Hz and maximal intensity around 100 Hz". The mice's hearing sensitivity is between 1K-100kHz. Thus the maximum intensity frequency of 100Hz would still be infrasound. It would be essential to compare to humans because mice experienced this music with less intensity than humans.

When authors discuss the possible role of epinephrine in the BBB opening, it is important to mention the role of corticosterone as another stress hormone. It has been shown that corticosterone has the opposite effect on BBB and tighten BBB. Thus, it could be that both hormones contribute to regulation of BBB during stress, in this case, loud music.

Minor comments:

Add to the manuscript that you used fluorescein isothiocyanate (FITC) to assess permeability to macromolecules.

Inconsistent usage of the abbreviation opening of BBB, sometimes it is OBBB and sometimes BBBO.

In Fig.1 The Ki-maps from the same mice at different time points. Is it superimposed maps from all mice or one example mouse?

Page 9, line 5. "4h (histological data)". I believe it should be for 4 weeks.

Suppl material, table 1. Data for 90dB. n=11, legend said n = 10. Also, it is not a clear # of 4 animals. In 4 mice, BBB did not open after 4 hours or after 1 hour?

In supplemental material, add the description to videos.

Page 1. Correct typo in " All procedures were performed in accordance with the "Guide for the Care and Use of Laboratory Animals" (157). The experimental protocols were approved by the Local Bioethics Commission of the Saratov State University (Protocol No. 7); the Institutional Animal Care and Use Committee of the University of New Mexico, USA (#200247).

Referee: 3

Comments to the Author(s).

Semyachkina-Glushkovskaya et al. provide a putative mechanistic explanation behind the use of rock music to affect blood-brain barrier permeability (BBB-p) in mice. They used Scorpions' "Still Loving You" at 70 dB, 90 dB, and 100dB at .25hr, 1hr, and 2hr time points, to determine BBB permeability to Evans Blue Albumin Complex (EBAC, 68.5 kDa) between 1-4hrs after exposure. This work is most innovative, and there is practically the only team that is addressing such questions. They have a history of rigorous research.

In and Ex vivo real-time fluorescent microscopy analysis confirmed highest concentrations of EBAC within cerebral microvessels after 1hr exposure to music (100 dB, 11-10,000 Hz, 2h intermittent mode: 60 sec – music; 60 sec – pause). A second analysis of BBB-p involved ex vivo confocal microscopy on brain slices and in vivo two-photon laser scanning microscopy (2PLSM). Ex vivo confocal microscopy revealed the accumulation of FITCD outside of brain capillaries 1h after music exposure. No leakage of FITCD was observed in the control group (no music). The leakage of FITCD in 2PLSM was also observed 1h after music exposure and was quantified by measuring the percentage fluorescence intensity of FITCD in the perivascular area. A third analysis of BBB-p involved magnetic resonance imaging (MRI) of gadolinium-diethylenetriamine-pentaacetic acid (Gd-DTPA) leakage. Results indicate a statistically significant ($p < .001$) increase in the BBB permeability to Gd-DTPA in all regions of the brain 1h after music exposure. Altogether, the results demonstrate that loud music exposure (90-100 dB, 11 – 10,000 Hz, Scorpions "Still loving you") during 2 hrs in intermittent adaptive mode (1 min – sound; 1 min – pause) is accompanied by delayed (1h after music impact) and short-lasting (during 1-4 hrs) OBBB to low and high molecular weight compounds.

The stress response to loud music involved the release of epinephrine, as confirmed by an increase in the plasma level of epinephrine up to 3.1-fold (immediately after the music) vs. the normal state (9.0 ± 1.5 ng/ml vs. 2.9 ± 0.7 ng/ml, $p < .001$, $n=10$ in each group). One hour after music exposure (the time of OBBB), the level of epinephrine returned almost to the normal value of 3.9 ± 1.6 ng/ml ($n=10$) and was over the control units 4h after music impact (the time of BBB closing) (2.5 ± 0.1 ng/ml, $n=10$). To test this, they measured changes in cerebral blood flow at the venous (the Sagittal sinus) and microcirculatory levels in the same 10 mice before, immediately, 1h, 4h, and 24h after sound exposure. The results demonstrate that immediately after music rCBF was increased in both venous and microcirculatory levels compared with the control group (0.80 ± 0.03 a.u. vs. 0.58 ± 0.01 a.u., $p < .001$ for the cerebral microvessels; 1.22 ± 0.01 a.u. vs. 0.83 ± 0.02 a.u., $p < .001$ for the Sagittal sinus). One hour after sound exposure, when the BBB was opened, rCBF tended to decrease but continued to be higher than the normal level of rCBF (0.77 ± 0.08 a.u. vs. 0.58 ± 0.01 a.u., $p < .05$ for the cerebral microvessels; 0.92 ± 0.07 a.u. vs. 0.83 ± 0.02 a.u., $p < .05$ for the Sagittal sinus).

Finally, they studied integrity of tight junction proteins (claudin-5 (CLDN-5), occluding (OCC) and zonula occludens (ZO-1)) immediately after music effects, during OBBB (1h after the music), and in the time of BBB recovery (4h and 24h after the music) compared with the control group (no music). Results show that the signal intensity from CLDN-5 and OCC but not from ZO-1 were significantly decreased in the time of OBBB. There were no changes in the TJ assembly immediately after listening to music as well as in the time of BBB restoration (4h and 24 after the music). Importantly, music-OBBB for FITCD was accompanied by lymphatic clearance of FITCD from the brain with its accumulation in the enlarged lymphatic vessels of dcLNs (deep cervical lymph nodes) that was not observed in intact mice. Using confocal imaging of the deep cervical lymph nodes (dcLNs), which are the first anatomical station of the cerebral spinal fluid (CSF) outflow, they show that music-induced OBBB for FITCD was accompanied by FITCD lymphatic clearance from the brain with its accumulation in the enlarged lymphatic vessels of dcLNs (22.3 ± 1.5 μ m vs. 37.3 ± 2.0 μ m, $p < .001$) that was not observed in intact mice.

The limitation section could be expanded. For example, variability in the potential effects of the complexity of the music used in this study may have skewed results. The musical selection used in this study was Scorpions' "Still Loving You." The authors tested varying loudness levels in decibels (70, 90, and 100), varying durations (.25hr, 1hr, and 2hr), and varying deliveries (60sec music – 60sec pause). Comparison/contrast of effect(s) upon BBB-p of other musical

stimuli/selections was not conducted. Such a comparison/contrast may have concretely concluded that volume level is indeed the sole reason for BBB-p, rather than effects of other musical elements (e.g., melody, timbre, harmony, rhythm, tempo, texture, form, etc.), and/or combination of volume with other musical elements upon BBB-p. Inclusion of comparator musical piece(s)/song(s) involving distinctly different musical qualities to "Still Loving You" would provide complementary evidence of the efficacy of S-G et al. (2020) use of this song to draw their conclusions. Future work should include a piece(s) of the music of distinctly different tempo/rhythm, key signature/mode, instrumentation/timbre/arrangement, and/or texture (e.g., homophony, heterophony, etc.). Furthermore, instrumental vs. vocal vs. instrumental/vocal combinations could be tested to concretely exclude these variables.

Author's Response to Decision Letter for (RSPB-2020-2337.R0)

See Appendices D & E.

Decision letter (RSPB-2020-2337.R1)

18-Nov-2020

Dear Dr Semachkina-Glushkovskaya

I am pleased to inform you that your manuscript entitled "Phenomenon of music-induced opening of the blood-brain barrier in healthy mice" has been accepted for publication in Proceedings B.

Your article has been estimated as being 9 pages long. Our Production Office will be able to confirm the exact length at proof stage.

Open Access

Paper charges

Sincerely,
Editor, Proceedings B
mailto: proceedingsb@royalsociety.org

Appendix A

A major challenge of targeted drug delivery to the brain is the lack of permeability to large molecules afforded by the blood-brain barrier (BBB). The authors of this study are demonstrating the ability of sound and music played at moderate-to-high intensity levels for different periods of time to induce a temporary increase in permeability to high molecular weight molecules, in a manner that is reversible. To achieve this, the authors exposed male mice to varying levels of sound/music through loudspeakers followed by injection with a dye that binds serum albumin ("EBAC") to assess BBB permeability. The authors used a sophisticated confocal live-imaging paradigm to monitor fluorescence intensity of the sound/music-induced changes in BBB permeability in conjunction with two photon imaging and MRI, along with biochemical analysis to assess the efficacy of this method. The major finding of these studies is that 2 h of exposure to 90-100 dB produces a significant increase in permeability of the BBB, as measured by levels of EBAC, and that levels go back to baseline hours after stimulation. Whilst these results serve as a good proof of concept for exploration of sound/music-induced targeted drug delivery to the brain for different neurodegenerative diseases or trauma to the brain, there are some major issues that the authors must address that I have broken into two broad topics below, that are not mutually exclusive.

The first issue to address is with regards to the specificity and efficacy of the drug delivery/BBB breakdown method shown in this study. In the introduction the authors introduce the lack of drug delivery methods to the brain that are non-invasive, but spatiotemporally specific and allowing of high concentration of given molecules that are to be delivered. The authors do not discuss how the current method provides solutions to these shortcomings. For example, after the drug has been administered in a mouse, does keeping them in sound isolation improve the sustained concentration of the drug for longer periods of time? The effect of different levels of noise post-stimulation on the pharmacokinetics of the delivered drug seems like an important variable to take into account, as there may be a threshold past which the efficacy declines. Specific, targeted delivery of drugs in different brain regions should also be addressed. The authors should also comment on why they chose the frequency of 370 Hz to use in their non-music sound stimulation, as opposed to a series of pure tones of different frequencies, which seems like the more obvious approach to initially explore.

The second, and possibly more significant issue the authors should address is the high probability of unintended damage to cochlear synapses that can be caused by the high sound levels that are being used to achieve the robust result. The authors do address sensorineural hearing loss, however, they specifically compare their sound-stimulated mice to mice that are given *profound* deafness via high dose of aminoglycoside antibiotics that kills the sensory hair cells of the cochlea and show a permanent shift in auditory thresholds and auditory brainstem response. This however, does not seem like an appropriate control, since the sound stimulation used by the authors (2h of 90-100 dB) is virtually identically to the sound paradigm used to cause synapse damage, but not loss of sensory hair cells, described by Kujawa and Liberman (2h 100 dB SPL) resulting in noise-induced cochlear "synaptopathy" (NICS). Importantly, this noise paradigm results in a) a 25-30% reduction of cochlear synapse counts, b) a temporary shift in auditory thresholds, and c) auditory brainstem response wave I recovery of amplitude 2 weeks after noise exposure (Kujawa & Liberman, 2009). In order to address this, I would recommend for the authors to assess the extent to which the sound/music treatment causes NICS, as it is

now well-established that such hearing loss contributes to higher probability of onset of dementia (Fattal, Hansen, & Fritzsich, 2018).

Other comments:

Data availability: It states that the author has no data to provide, why is this and is it an issue that concerns the current review stage?

There are a lot of methods here, and there should be a lot of data and figures to go with them. In the current manuscript there does not appear to be enough figures or data to determine a lot of the results that have been stated. It appears that there are references to figures (e.g. Figure 3) that there is not corresponding figure for.

Fattal, D., Hansen, M., & Fritzsich, B. (2018). Aging-Related Balance Impairment and Hearing Loss. *The Wiley handbook on the aging mind and brain*, 315-336.

Kujawa, S. G., & Liberman, M. C. (2009). Adding insult to injury: cochlear nerve degeneration after "temporary" noise-induced hearing loss. *The Journal of Neuroscience*, 29(45), 14077-14085. Retrieved from <http://www.jneurosci.org/cgi/content/full/29/45/14077>

Appendix B

Comments: A major challenge of targeted drug delivery to the brain is the lack of permeability to large molecules afforded by the blood-brain barrier (BBB). The authors of this study are demonstrating the ability of sound and music played at moderate-to-high intensity levels for different periods of time to induce a temporary increase in permeability to high molecular weight molecules, in a manner that is reversible. To achieve this, the authors exposed male mice to varying levels of sound/music through loudspeakers followed by injection with a dye that binds serum albumin (“EBAC”) to assess BBB permeability. The authors used a sophisticated confocal live-imaging paradigm to monitor fluorescence intensity of the sound/music-induced changes in BBB permeability in conjunction with two photon imaging and MRI, along with biochemical analysis to assess the efficacy of this method. The major finding of these studies is that 2 h of exposure to 90-100 dB produces a significant increase in permeability of the BBB, as measured by levels of EBAC, and that levels go back to baseline hours after stimulation. Whilst these results serve as a good proof of concept for exploration of sound/music-induced targeted drug delivery to the brain for different neurodegenerative diseases or trauma to the brain, there are some major issues that the authors must address that I have broken into two broad topics below, that are not mutually exclusive. The first issue to address is with regards to the specificity and efficacy of the drug delivery/BBB breakdown method shown in this study. In the introduction the authors introduce the lack of drug delivery methods to the brain that are non-invasive, but spatiotemporally specific and allowing of high concentration of given molecules that are to be delivered. The authors do not discuss how the current method provides solutions to these shortcomings. For example, after the drug has been administered in a mouse, does keeping them in sound isolation improve the sustained concentration of the drug for longer periods of time? The effect of different levels of noise post-stimulation on the pharmacokinetics of the delivered drug seems like an important variable to take into account, as there may be a threshold past which the efficacy declines. Specific, targeted delivery of drugs in different brain regions should also be addressed. The authors should also comment on why they chose the frequency of 370 Hz to use in their non-music sound stimulation, as opposed to a series of pure tones of different frequencies, which seems like the more obvious approach to initially explore. The second, and possibly more significant issue the authors should address is the high probability of unintended damage to cochlear synapses that can be caused by the high sound levels that are being used to achieve the robust result. The authors do address sensorineural hearing loss, however, they specifically compare their sound-stimulated mice to mice that are given profound deafness via high dose of aminoglycoside antibiotics that kills the sensory hair cells of the cochlea and show a permanent shift in auditory thresholds and auditory brainstem response. This however, does not seem like an appropriate control, since the sound stimulation used by the authors (2h of 90-100 dB) is virtually identically to the sound paradigm used to cause synapse damage, but not loss of sensory hair cells, described by Kujawa and Liberman (2h 100 dB SPL) resulting in noise-induced cochlear “synaptopathy” (NICS). Importantly, this noise paradigm results in a) a 25-30% reduction of cochlear synapse counts, b) a temporary shift in auditory thresholds, and c) auditory brainstem response wave I recovery of amplitude 2 weeks after noise exposure (Kujawa & Liberman, 2009). In order to address this, I would recommend for the authors to assess the extent to which the sound/music treatment causes NICS, as it is now well-established that such hearing loss contributes to higher probability of onset of dementia (Fattal, Hansen, & Fritzsche, 2018). Other comments: Data availability: It states that the author has no data to provide, why is this and is it an issue that concerns the current review stage? There are a lot of methods here, and there should be a lot of data and figures to go with them. In the current manuscript there does not appear to be enough figures or data to determine a lot of the results that have been stated. It appears that there are references to figures (e.g. Figure 3) that there is not corresponding figure for.

Response: We would like to express our gratitude to Reviewer for helpful advices and constructive comments. We changed our manuscript completely. Due to limitation of paper size (6 pages) by the journal “Proceedings of The Royal Society B” in the previous version of our manuscript we could not show all results of our research. We also understood not correctly the format of journal and decided that we cannot show supplementary information (SI); now we added SI. The current version of paper is focused on the study of loud music effects on BBB. We removed the results, which demonstrate infrasound effects on the BBB integrity because we submitted patent based on infrasound-mediated brain drug delivery in humans and now we cannot publish these data until we will receive permission to do it.

In our previous work, we found that loud music has similar effects with infrasound. Music is an important part of our life and millions of teenagers and professional rock, jazz and symphony orchestra musicians are in a potential risk group of opening of BBB (OBBB) due to exposure to loud music in concerts, nightclubs, discotheques, and pubs. Therefore, we expect that our results demonstrating music effects on BBB will be interesting for readers. We clearly demonstrate that the listening of loud music during 2 hrs in an intermittent adaptive regime is accompanied by delayed (1h after music exposure) and short-lasting (during 1-4 hrs) OBBB to low and high molecular weight compounds without cochlear and brain impairments. We present the systemic and molecular mechanisms responsible for music-induced OBBB. Finally, a revision of our classic knowledge about the BBB nature and the novel strategies in optimization of sound-mediated methods for brain drug delivery are discussed.

We chose the intermittent music treatment (1 min – sound; 1 min – pause) because the safe listening time for continues sound of 100 dB is 15 min (<https://www.hear-it.org/disco-noise-volume-over-the-top-1>). However, we found no effect on BBB when mice listened continues music of 100 dB during 15 min (Table in SI). Therefore, the longer music exposure during 1h and 2 h in repetitive mode was chosen for adaptation to loud sound. When stimuli are repeated, neural activity is usually reduced due to adaptation (doi.org/10.1016/j.tics.2005.11.006). This neural repetition effects have been reported at multiple spatial scales from the level of individual cortical neurons to the level of hemodynamic changes as adaptive mechanisms of the nervous system ([doi.org/10.1016/0006-8993\(94\)90061-2](https://doi.org/10.1016/0006-8993(94)90061-2); [doi.org/10.1016/0166-4328\(95\)00197-2](https://doi.org/10.1016/0166-4328(95)00197-2); [doi.org/10.1016/S0001-6918\(01\)00019-1](https://doi.org/10.1016/S0001-6918(01)00019-1)). We determined that a stimulus period of 2 h at 90-100 dB produced significant OBBB. No changes in the BBB permeability were found when the stimulus was 70 dB or when 1h music exposure was used.

Using 2h intermittent music exposure, we found no appearance of apoptotic cells, morphological alterations in the brain tissues, or in the spiral ganglion cell (SGCs) 1h after music impact when the BBB was opened and 4h (histological data) after music influences when the BBB integrity was fully restored, or 4 weeks after delayed music effects. We hypothesize that the listening of loud music in the repetitive mode can adapt the brain to sound stress and protect its structure from sound-induced injuries.

We performed additional experiments following advices of Reviewer and demonstrated in session “*Mechanisms underlying music-induced OBBB*” the effects of loud music on the cochlear as well as the changes in the serum epinephrine level, rCBF, and TJ assembly immediately, 1h, 4h and 24h after music exposure. We also discuss the short and long-term effects (2 and 4 weeks) of loud music on the brain. Additionally, we performed experiments to study the time window of OBBB (Table 1 in SI) and the brain regions with music-induced OBBB (Figure 3 in SI).

We would like to thanks again Reviewer for constructive comments and advices.

Authors

Appendix C

Reviewer 1

Comments: The authors investigated the effect of music/sound on the blood-brain barrier using a mouse model. They found that two hours of 90-100 dB music/sound safely open the BBB to low/high molecular weight molecules in mice. The authors argue that BBB opens after one hour, and closes after 4 or 24 hours. The results of this study could have a potential impact on non-invasive therapy for drug delivery in humans. The authors used a rigorous analysis to address the mechanism of BBB opening as a result of sound exposure. I have comments and suggestions aimed to address some conceptual problems derived from the experimental approach and also sought to improve the presentation of the paper. In this study, the authors argue that both music and sound open BBB. Except for results regarding permeability to the Evans Blue Albumin Complex, all data in the figures are shown for the sound. Music of 90-100 dB (11-10,000 Hz) and sound of 90-100 dB (370Hz) are different stimuli since the mice hearing sensitivity is between 1K-100kHz vs. human's one (100Hz-20kHz). Thus, the sound (370Hz) used in this study is infrasound for mice. What was the reason for authors to chose the infrasound as a stimulus in the experiments? What changes this sound produce in cochlea? The fact that authors used infrasound in this study and observed the opening of BBB could suggest a different mechanism of opening of BBB, for example, via mechanoreceptors. There are studies showing that infrasound affects the outer hair cells but not inner hair cells. This should be addressed in the discussion. The authors should present data for music as well and compare those to sound one. The authors argue that "the BBB disruption caused by sound is associated with a post-stress period (1h after sound exposure) when the serum epinephrine level was significantly decreased compared with stress". (please, correct typo in "epinephrine", "decreased"). However, they tested BBB in sound experiments after one hour, not immediately after. The rise in epinephrine induced by stress is known to increase cerebral blood flow and to cause leaks in the BBB. Is it possible that the BBB opening was initiated earlier and could be explained, at least partially, by the rise of epinephrine? In support, their data showed that immediately after the sound, "the CBF was increased in both venous and microcirculatory levels" (fig 2b). The authors argue that this approach is safe and did not produce hearing problems in mice. They used the fear conditioning test to assess the hearing status. Many studies are suggesting that sound above 85dB could be damaging for the hair cells. The fear conditioning test does not reflect changes in hair cells of the cochlea. Also, the authors argue that this could be used in humans. Two hours of 90-100 dB music/sound exposure is quite a long time, and potentially, it could be harmful. Authors should be cautious when they discuss it. The manuscript contains multiply typos and errors that need to be corrected. Page 2. In Author-supplied statements. Data. To question, "It is a condition of publication that data, code, and materials supporting your paper are made publicly available. Does your paper present new data?" The authors wrote, "My paper has no data." Please, explain. Page. 7. Introduction. After "focused ultrasound," add abbreviation because later authors refer to FUS, and it could be confusing. Page 7. Introduction. Authors wrote, "There are no special centers, which are involved in the development of AD or ASL. Therefore, the brain region for therapy of these diseases and targeted FUS-related brain drug delivery is not clear. " It is a confusing statement because many research centers are focusing on studying Alzheimer's disease and brain regions that are involved in developing this disease. The authors need to elaborate on their statement. The word "involved" misspelled.

Response: We would like to express our gratitude to Reviewer for helpful advices and constructive comments. We changed our manuscript completely. Due to limitation of paper size (6 pages) by the journal "Proceedings of The Royal Society B" in the previous version of our manuscript we could not show all results of our research. We also understood not correctly the format of the journal and decided that we cannot show supplementary information (SI); now we added SI. The

current version of paper is focused on the study of loud music effects on BBB. We removed the results, which demonstrate infrasound effects on the BBB integrity because we submitted patent based on infrasound-mediated brain drug delivery in humans and now we cannot publish the data until we will receive permission to do it. In previous version of our paper, we did not have idea to make comparison loud music and infrasound. We demonstrated that the BBB opening (OBBB) depends on the intensity of sound but not on its frequency. Therefore, we showed the similar effects of music and infrasound on BBB using spectrofluorometric assay of Evans Blue leakage. Afterward, sound effects on the BBB permeability to low and high weight compounds were included only to show that sound itself without microbubbles (as FUS) or any other interventions can open BBB.

In our work, we found that loud music has similar effects with infrasound. Music is an important part of our life and millions of teenagers and professional rock, jazz and symphony orchestra musicians are in a potential risk group of OBBB due to exposure to loud music in concerts, nightclubs, discotheques, and pubs. Therefore, we expect that our results demonstrating music effects on BBB will be interesting for readers. We clearly demonstrate that the listening of loud music during 2 hrs in an intermittent adaptive regime is accompanied by delayed (1h after music exposure) and short-lasting (during 1-4 hrs) OBBB to low and high molecular weight compounds without cochlear and brain impairments. We present the systemic and molecular mechanisms responsible for music-induced OBBB. Finally, a revision of our classic knowledge about the BBB nature and the novel strategies in optimization of sound-mediated methods for brain drug delivery are discussed.

We chose the intermittent music treatment (1 min – sound; 1 min – pause) because the safe listening time for continuous sound of 100 dB is 15 min (<https://www.hear-it.org/disco-noise-volume-over-the-top-1>). However, we found no effect on BBB when mice listened continuous music of 100 dB during 15 min (Table in SI). Therefore, the longer music exposure during 1h and 2 h in repetitive mode has been chosen for adaptation to loud sound. When stimuli are repeated, neural activity is usually reduced due to adaptation (doi.org/10.1016/j.tics.2005.11.006). This neural repetition effects have been reported at multiple spatial scales from the level of individual cortical neurons to the level of hemodynamic changes as adaptive mechanisms of the nervous system ([doi.org/10.1016/0006-8993\(94\)90061-2](https://doi.org/10.1016/0006-8993(94)90061-2); [doi.org/10.1016/0166-4328\(95\)00197-2](https://doi.org/10.1016/0166-4328(95)00197-2); [doi.org/10.1016/S0001-6918\(01\)00019-1](https://doi.org/10.1016/S0001-6918(01)00019-1)). We determined that a stimulus period of 2 h at 90-100 dB produced significant OBBB. No changes in the BBB permeability were found when the stimulus was 70 dB or when 1h music exposure was used.

Using 2h intermittent music exposure, we found no appearance of apoptotic cells, morphological alterations in the brain tissues, or in the spiral ganglion cell (SGCs) 1h after music impact when BBB was opened and 4h (histological data) after music influences when the BBB integrity was fully restored, or 4 weeks after delayed music effects. We hypothesize that the listening of loud music in the repetitive mode can adapt the brain to sound stress and protect its structure from sound-induced injuries.

We performed additional experiments and demonstrated in session “*Mechanisms underlying music-induced OBBB*” the effects of loud music on the cochlear as well as the changes in the serum epinephrine level, rCBF, and TJ assembly immediately, 1h, 4h and 24h after music exposure. We also discuss the short and long-term effects (2 and 4 weeks) of loud music on the brain. Additionally, we performed experiments to study the time window of OBBB (Table 1 in SI) and the brain regions with music-induced OBBB (Figure 3 in SI).

Comment: Page 14. Methods. "Evans Blue dye (Sigma Chemical Co., St. Louis, Michigan". Sigma Chemical Co is located in St. Louis, Missouri. Please, correct it.

Response: We corrected it.

Comment: Page 16. Methods. In Assessment of apoptosis with TUNEL method. "The number of apoptotic cells before and in 1h after sound exposure (n=7 in each group) was evaluated..". Please, explain why this analysis was conducted only in 1h after sound exposure.

Response: We added the results of the short-term and delayed effects of loud music on the brain and the auditory system.

“To analyze the short-term and delayed effects of loud music on the brain and auditory system injury, whole brain and the cochlear histologic examination was performed using haematoxylin and eosin staining and TUNEL staining for apoptosis 1h (the time of OBBB), 4h (the time of BBB closing), and 4 weeks (delayed effects) after music exposure. The timing of the experiments was dictated by the fact that sound damage may not appear immediately, but after a certain time. For example, peripheral synaptic connections of cochlear neurons are the most vulnerable elements in the cochlea and, in the vast majority of cases, cochlear nerve fibers degenerate only long time after the sound trauma [31]. In animal models exposed to sound stress, hair cell loss can be widespread within hours [31-35], whereas the loss of the spiral ganglion cell (SGCs) is typically not detectable for several weeks to month after sound exposure [31,34,35]. The apoptosis evaluation after ultrasound-OBBB usually performed for several hours to 4-5 weeks after OBBB [36, 37]. Therefore, the time of analysis of SGCs and apoptosis was chosen 1h (the time of OBBB) and 4 weeks (delayed effects) after music exposure.

Figures 2 b-e present cochlear histology and quantification of SGNs density in intact mice (no music), 1h and 4 weeks after music exposure. Our results did not reveal any morphological changes in SGCs after short and delayed effects of loud music on the cochlear system. We found that loud music did not induce apoptosis in the mouse brain tissues (data not presented). A histological analysis of brain tissues and the cerebral vessels also did not show any sound-mediated injuries 1h, 4h and 4 weeks after music exposure (Figure 5 in SI)”.

Comment: Page 8. Results. Authors wrote, "Thus, this series of experiments demonstrated that loud music/sound (100 dB) reversibly opens the BBB to EBAC in all tested mice and these changes are similar either for music or sound". This conclusion is confusing since authors found that only "stimulus period of 2 h at 90-100 dB produced a robust increase in the BBB", while 70 dB was ineffective and "4h and 24h after either type of stimulation the BBB permeability to EBAC had returned to levels near the normal state..". The authors should be more specific and accurate in the description of the results. Page Page 8. Results. "Figures 1, o,n show post sound the BBB disruption.." There is no Figure 1 o. Page. 10. In Results. "In the first step, we answered the question, what is the role of stress in sound-induced BBB opening. The loud sound caused an increase in the plasma level of important stress hormone such as epinephrine up to 7.0-fold (immediately after sound) vs. the normal state (23.1 ± 2.7 ng/ml vs. 3.3 ± 0.9 ng/ml, $p < 0.001$, $n = 10$ in each group)". The authors need to clarify to the time of the measurement and how long the music/sound was delivered (0.25, 1 h or 2h).

Response: We considered these shortcomings in the new version of manuscript.

Comment: Page. 10. results. The "audible system" needs to be correct to the "auditory system" throughout the text. Page 13. Discussion, "acoustically", correct the typo.

Response: We corrected it.

Comment: Figure 2 has a poor legend description, lack of abbreviation, errors. "e - The schema of mechanisms of sound-induced BBB opening". It should be "d" and correct to "schematic illustration of...". Add all necessary abbreviations, and it is not clear to what "before" and "After" sound is related to what figure? In Figure 2b, define blue and red color.

Response: We added new figures in the novel version of manuscript.

Comment: Page 11. The authors wrote, "Our data are consistent with the other results suggesting an increase in CBF via dilation of the cerebral vessels in humans and in the cochlear blood flow in the guinea pig after the electrical stimulation of the vestibular nerve [13,14]". This is a very confusing and misleading conclusion since this manuscript does not study the vestibular activation at all. The study [ref. 13] addresses precisely how the stimulation of the vestibular system by tilting and translating subject changes the cerebral blood flow. The authors in ref 14 studied cochlear blood flow in the guinea pig by stimulating different parts of the cochlea and cervical ganglion.

Response: We corrected it.

We would like to thanks again Reviewer for constructive comments and advices.

Authors

Reviewer 2

Comments: A major challenge of targeted drug delivery to the brain is the lack of permeability to large molecules afforded by the blood-brain barrier (BBB). The authors of this study are demonstrating the ability of sound and music played at moderate-to-high intensity levels for different periods of time to induce a temporary increase in permeability to high molecular weight molecules, in a manner that is reversible. To achieve this, the authors exposed male mice to varying levels of sound/music through loudspeakers followed by injection with a dye that binds serum albumin (“EBAC”) to assess BBB permeability. The authors used a sophisticated confocal live-imaging paradigm to monitor fluorescence intensity of the sound/music-induced changes in BBB permeability in conjunction with two photon imaging and MRI, along with biochemical analysis to assess the efficacy of this method. The major finding of these studies is that 2 h of exposure to 90-100 dB produces a significant increase in permeability of the BBB, as measured by levels of EBAC, and that levels go back to baseline hours after stimulation. Whilst these results serve as a good proof of concept for exploration of sound/music-induced targeted drug delivery to the brain for different neurodegenerative diseases or trauma to the brain, there are some major issues that the authors must address that I have broken into two broad topics below, that are not mutually exclusive. The first issue to address is with regards to the specificity and efficacy of the drug delivery/BBB breakdown method shown in this study. In the introduction the authors introduce the lack of drug delivery methods to the brain that are non-invasive, but spatiotemporally specific and allowing of high concentration of given molecules that are to be delivered. The authors do not discuss how the current method provides solutions to these shortcomings. For example, after the drug has been administered in a mouse, does keeping them in sound isolation improve the sustained concentration of the drug for longer periods of time? The effect of different levels of noise post-stimulation on the pharmacokinetics of the delivered drug seems like an important variable to take into account, as there may be a threshold past which the efficacy declines. Specific, targeted delivery of drugs in different brain regions should also be addressed. The authors should also comment on why they chose the frequency of 370 Hz to use in their non-music sound stimulation, as opposed to a series of pure tones of different frequencies, which seems like the more obvious approach to initially explore. The second, and possibly more significant issue the authors should address is the high probability of unintended damage to cochlear synapses that can be caused by the high sound levels that are being used to achieve the robust result. The authors do address sensorineural hearing loss, however, they specifically compare their sound-stimulated mice to mice that are given profound deafness via high dose of aminoglycoside antibiotics that kills the sensory hair cells of the cochlea and show a permanent shift in auditory thresholds and auditory brainstem response. This however, does not seem like an appropriate control, since the sound stimulation used by the authors (2h of 90-100 dB) is virtually identically to the sound paradigm used to cause synapse damage, but not loss of sensory hair cells, described by Kujawa and Liberman (2h 100 dB SPL) resulting in noise-induced cochlear “synaptopathy” (NICS). Importantly, this noise paradigm results in a) a 25-30% reduction of cochlear synapse counts, b) a temporary shift in auditory thresholds, and c) auditory brainstem response wave I recovery of amplitude 2 weeks after noise exposure (Kujawa & Liberman, 2009). In order to address this, I would recommend for the authors to assess the extent to which the sound/music treatment causes NICS, as it is now well-established that such hearing loss contributes to higher probability of onset of dementia (Fattal, Hansen, & Fritzsche, 2018). Other comments: Data availability: It states that the author has no data to provide, why is this and is it an issue that concerns the current review stage? There are a lot of methods here, and there should be a lot of data and figures to go with them. In the current manuscript there does not appear to be enough figures or data to determine a

lot of the results that have been stated. It appears that there are references to figures (e.g. Figure 3) that there is not corresponding figure for.

Response: We would like to express our gratitude to Reviewer for helpful advices and constructive comments. We changed our manuscript completely. Due to limitation of paper size (6 pages) by the journal “Proceedings of The Royal Society B” in the previous version of our manuscript we could not show all results of our research. We also understood not correctly the format of journal and decided that we cannot show supplementary information (SI); now we added SI. The current version of paper is focused on the study of loud music effects on BBB. We removed the results, which demonstrate infrasound effects on the BBB integrity because we submitted patent based on infrasound-mediated brain drug delivery in humans and now we cannot publish these data until we will receive permission to do it.

In our previous work, we found that loud music has similar effects with infrasound. Music is an important part of our life and millions of teenagers and professional rock, jazz and symphony orchestra musicians are in a potential risk group of opening of BBB (OBBB) due to exposure to loud music in concerts, nightclubs, discotheques, and pubs. Therefore, we expect that our results demonstrating music effects on BBB will be interesting for readers. We clearly demonstrate that the listening of loud music during 2 hrs in an intermittent adaptive regime is accompanied by delayed (1h after music exposure) and short-lasting (during 1-4 hrs) OBBB to low and high molecular weight compounds without cochlear and brain impairments. We present the systemic and molecular mechanisms responsible for music-induced OBBB. Finally, a revision of our classic knowledge about the BBB nature and the novel strategies in optimization of sound-mediated methods for brain drug delivery are discussed.

We chose the intermittent music treatment (1 min – sound; 1 min – pause) because the safe listening time for continuous sound of 100 dB is 15 min (<https://www.hear-it.org/disco-noise-volume-over-the-top-1>). However, we found no effect on BBB when mice listened continuous music of 100 dB during 15 min (Table in SI). Therefore, the longer music exposure during 1h and 2 h in repetitive mode was chosen for adaptation to loud sound. When stimuli are repeated, neural activity is usually reduced due to adaptation (doi.org/10.1016/j.tics.2005.11.006). This neural repetition effects have been reported at multiple spatial scales from the level of individual cortical neurons to the level of hemodynamic changes as adaptive mechanisms of the nervous system ([doi.org/10.1016/0006-8993\(94\)90061-2](https://doi.org/10.1016/0006-8993(94)90061-2); [doi.org/10.1016/0166-4328\(95\)00197-2](https://doi.org/10.1016/0166-4328(95)00197-2); [doi.org/10.1016/S0001-6918\(01\)00019-1](https://doi.org/10.1016/S0001-6918(01)00019-1)). We determined that a stimulus period of 2 h at 90-100 dB produced significant OBBB. No changes in the BBB permeability were found when the stimulus was 70 dB or when 1h music exposure was used.

Using 2h intermittent music exposure, we found no appearance of apoptotic cells, morphological alterations in the brain tissues, or in the spiral ganglion cell (SGCs) 1h after music impact when the BBB was opened and 4h (histological data) after music influences when the BBB integrity was fully restored, or 4 weeks after delayed music effects. We hypothesize that the listening of loud music in the repetitive mode can adapt the brain to sound stress and protect its structure from sound-induced injuries.

We performed additional experiments following advices of Reviewer and demonstrated in session “*Mechanisms underlying music-induced OBBB*” the effects of loud music on the cochlear as well as the changes in the serum epinephrine level, rCBF, and TJ assembly immediately, 1h, 4h and 24h after music exposure. We also discuss the short and long-term effects (2 and 4 weeks) of

loud music on the brain. Additionally, we performed experiments to study the time window of OBBB (Table 1 in SI) and the brain regions with music-induced OBBB (Figure 3 in SI).

We would like to thanks again Reviewer for constructive comments and advices.

Author

Appendix D

Comments: This is a revised version of the manuscript previously titled "Music/sound opens the blood-brain barrier: a readily available approach to brain drug delivery and therapy of brain diseases." The authors changed the presentation leaving only data regarding the effect of music on the BBB. The authors revised the manuscript's primary focus from presenting it as a search for a non-invasive approach for increasing BBB permeability to describing a phenomenon of BBB opening during the music in general. They carefully discussed this phenomenon as a potential usage in drug-delivery therapy at the end. I agree with these changes because more studies are needed to understand the BBB opening mechanisms during loud music. Loud music/sound could potentially negatively affect the hearing. Overall, the author substantially improved the manuscript in all aspects: discussion, methods, modified and added information in figures, and corrected many other issues.

My main comment is about the description of the main findings. Authors found that exposure to music 90-100 dB during 2 hours increases the BBB permeability and that significant changes in BBB were observed at 1 hour after music-off time. The authors wrote, "This method also revealed the delayed OBBB that was only 1-4 h after music effects but not immediately music-off". It is confusing because, at 4 hours after the music stopped, the BBB activity did not differ from the control group. Thus, based on the available data, the authors cannot specify how long the BBB stayed opened: 15 min, one hour, or 2-3 hours. For this, more data need to be collected at a different time (1,2,3, hours). If authors have additional data for time between 1-4 hours, then they could present it. If they do not have it, they need to state what they found precisely.

Response: We want to express our sincere gratitude for hard work with our manuscript and deep analysis of our results as well as for constructive and helpful advices. We added the additional data demonstrating that OBBB was only 1 h after music effects but not immediately or 15/30 min music-off. We also modified sentence with information about the closing of BBB. The recovery of BBB was individual in each mouse. The fluorescent imaging and two-photon microscopy do not allow to control the closing of BBB in real time because the fluorescent signal from the tested tracers remained strong during long time after its crossing the opened of BBB. We analyzed the recovery of BBB using the spectrofluorometric quantitative assay of EBAC extravasation in different groups. However, due to the high variability of units and wide recovery time range, we decided to show only one time when all mice demonstrated no BBB permeability to EBAC, it was 4 h after music exposure.

The window of music-induced OBBB

In the first step, we determined the optimal duration/intensity of music exposure for OBBB as well as the time window of OBBB after listening of loud music (the design of our experiments is presented in Figures 1-3 in SI). Song of the Scorpions "Still loving you" was administered for periods up to 0.25 (continues mode), 1 h and 2 h in an intermittent mode: 60 s – sound and 60 s – pause, at intensities ranging up to 70 dB (moderate), 90 dB (loud) and 100 dB (very loud) [20,21]. Using the *in vivo* real-time fluorescent microscopy [22] in awake behavior mice and optical clearing of skull [23], we determined that a stimulus period of 2 h at 90-100 dB produced a robust increase in the BBB permeability to the Evans Blue Albumin Complex (EBAC, 68.5 kDa) that was observed as bright intensity around the cerebral capillaries (Figure 1b). No changes in the BBB permeability were found in intact mice (no music, Figure 1a) or when the stimulus was 70 dB and shorter times of music exposure (0.25 h and 1 h) was used. This *in vivo* technique also revealed the delayed OBBB that was only 1 h after music effects but not immediately or 15/30 min music-off that also we confirmed in *ex vivo* experiments using spectrofluorometric quantitative assay of EBAC extravasation (Table 1 in SI).

These *ex vivo* findings also showed music-induced OBBB in 11 brain regions (Figure 3 in SI). The recovery of BBB was individual in each mouse with the high variability of units in the groups and wide recovery time range; however, there was no the permeability of BBB to EBAC 4 h after music exposure in all mice (Table 1 in SI). There was no OBBB in the next day after music exposure.

Table 1 SI. The effect of loud music on the BBB permeability to EBAC ($\mu\text{g/g}$ tissue)

Sound level (dB) and time elapsed after sound exposure (h)	Content of EBAC ($\mu\text{g/g}$ tissue)			
	No music (the control group)	Music duration during 2 h (intermittent)	Music duration during 1 h (intermittent)	Music duration during 0.25 h (continues)
		0.15 \pm 0.01		
100 dB				
immediately		0.12 \pm 0.01	0.11 \pm 0.05	0.14 \pm 0.02
¼ h		0.10 \pm 0.05	0.14 \pm 0.01	0.17 \pm 0.06
½ h		0.13 \pm 0.03	0.12 \pm 0.05	0.12 \pm 0.02
1 h		2.60 \pm 0.06 ***	0.12 \pm 0.03	0.17 \pm 0.01
4 h		0.19 \pm 0.03	0.15 \pm 0.07	0.15 \pm 0.05
24 h		0.16 \pm 0.03	0.11 \pm 0.09	0.17 \pm 0.02
90 dB				
immediately		0.15 \pm 0.08	0.11 \pm 0.06	0.17 \pm 0.01
1 h		2.70 \pm 0.04 ***(n=11)	0.16 \pm 0.03	0.16 \pm 0.08
4 h		0.18 \pm 0.06 (n=4)#	0.14 \pm 0.03	0.14 \pm 0.05
24 h		0.15 \pm 0.03	0.17 \pm 0.02	0.18 \pm 0.01
		0.19 \pm 0.07		
70 dB				
immediately		0.13 \pm 0.02	0.16 \pm 0.07	0.12 \pm 0.03
1 h		0.17 \pm 0.08	0.11 \pm 0.04	0.14 \pm 0.09
4 h		0.19 \pm 0.06	0.13 \pm 0.02	0.14 \pm 0.02
24 h		0.19 \pm 0.09	0.18 \pm 0.02	0.11 \pm 0.01

p<0.001: *** - vs. before music exposure (the control group), n=15 for the groups (music duration 2h) and n=10 for the groups (music duration 0.25-1 h); # - the number of mice without the BBB opening.

Comment: In Fig 2SI, the authors showed "the frequency range of music (Scorpions "Still Loving You"): frequencies in the range of 11-10,000 Hz and maximal intensity around 100 Hz". The mice's hearing sensitivity is between 1K-100kHz. Thus the maximum intensity frequency of 100Hz would still be infrasound. It would be essential to compare to humans because mice experienced this music with less intensity than humans.

Response: We added the discussion of the effects of infrasound on human brain.

"The frequencies range of music (Scorpions "Still Loving You") was 11-10,000 Hz and maximal intensity was around 100 Hz (Fig. 2 in SI). The mice's hearing sensitivity is between 1K - 90-100 kHz [42]. Thus, the maximum intensity frequency of 100 Hz for mice is infrasound (IS) for mice. The IS-induced opening of BBB has been shown in other experimental works [47,48]. However, there are no findings demonstrating direct IS effect on the BBB permeability in humans. There is a growing evidence that humans are indeed receptive to IS and that exposure to low-frequency sounds can give rise to high levels of annoyance, distress, sleep disturbances, headache and dizziness, over tinnitus and hyperacusis, to panic attacks and depression [49]. However, Leventhall [50] concluded that "if you cannot hear a sound and you cannot perceive it in other ways and it does not affect you". The World Health Organization suggests "there is no reliable evidence that ISs below the hearing threshold produce physiological or psychological effects" [51]. It appears that the notion, according to which sound needs to be perceived in order to exert

relevant effects on the organism, falls short when aiming at an objective risk assessment of IS, especially if one takes into consideration recent advances in research on inner ear physiology as well as on the effects of subliminal auditory stimulation (i.e. stimulation below the threshold of perception). So, 5-Hz IS exposure (60–65 dB) has been shown to trigger the response of inner ear components such as the outer hair cells in animals and it has been suggested that outer hair cell stimulation may also exert a broader influence on the nervous system via the brainstem [52]. In addition, there is the well documented effect in cognitive science that brain physiology and behavior can be influenced by a wide range of subliminally presented stimuli, including stimuli of the auditory domain [53]. The IS near the hearing threshold induces changes of neural activity across several brain regions, some of which are known to be involved in auditory processing, while others are regarded as keyplayers in emotional and vascular autonomic control in human [54]”.

42.Reynolds R.P., Kinard W.L., Degraff J.J., Leverage N., Norton J.N. 2010 Noise in a Laboratory Animal Facility from the Human and Mouse Perspectives. *JAALAS*, **49**(5), 592–597.

47.Fei, Z., Zhang, X., Wang, X., Li, Z., Lu, P., Liu, X., Chen, J., Jia, K. 2000 The changes of rat blood-brain barrier permeability and expression of heat shock protein 70 after infrasonic damage. *J. LOW. FREQ. NOISE. V. A.* **19**(2). (doi.org/10.1260/0263092001492840)

48.Liu, Z., Gong, L., Li, X., Ye, L., Wang, B., Liu, J., Qiu, J., Jiao H., Zhang W., Chen J., Wang J. 2012 Infrasound increases intracellular calcium concentration and induces apoptosis in hippocampi of adult rats. *Mol. Med. Rep.* **5**, 73-77. (doi: 10.3892/mmr.2011.597)

49.Shepherd D. McBride D., Welch D., Dirks K.N., Hill E.M. 2011 Evaluating the impact of wind turbine noise on health-related quality of life. *Noise Health.* **13**(54): 333–339. (doi: 10.4103/1463-1741.85502)

50.Leventhall HG. Low frequency noise and annoyance. 2004 *Noise Health.* 2004 **6**(23):59-72. <https://www.noiseandhealth.org/text.asp?2004/6/23/59/31663>

51.Berglund B, Lindvall T. Document prepared for WHO Archives of the Centre for Sensory Research. Stockholm University and Karolinska Institute; 1995:**2**: 1–195.

52.Salt A.N., Lichtenhan J.T., Gill R.M., Hartsock J.J. 2013 Large endolymphatic potentials from low-frequency and infrasonic tones in the guinea pig. *J Acoust Soc Am.* **133**(3): (doi: 1561–71. 10.1121/1.4789005)

53.Taylor E. *Thinking Without Thinking.* R K Book, Big Bear City, CA; 1994

54.Weichenberger M, Bauer M., Kühler R., Hensel J., Forlim C.G., Ihlenfeld A., Ittermann B., Gallinat J., Koch C., Kühn S., Zuo X. 2017 Altered cortical and subcortical connectivity due to infrasound administered near the hearing threshold – Evidence from fMRT. *PLoS One.* **12**(4): e0174420.

Comment: When authors discuss the possible role of epinephrine in the BBB opening, it is important to mention the role of corticosterone as another stress hormone. It has been shown that corticosterone has the opposite effect on BBB and tighten BBB. Thus, it could be that both hormones contribute to regulation of BBB during stress, in this case, loud music.

Response: We discuss the effects of corticosterone on the BBB permeability.

“We have present in detail possible systemic and molecular mechanisms responsible for music-induced OBBB. It is known that stress hormone, such as epinephrine, might induce an increase in the BBB permeability via vasodilation of cerebral vessels and an increase in the extension of cracks in the tight contacts of endothelial cells with a change in the ultrastructure of the astrocyte end-feet; increase transport and pinocytotic activity of endothelial cells [27-30]. The elevated stress hormones levels induced by sound were demonstrated in humans and animals [55]. Our results indicate that loud music induces a rise of serum epinephrine with a significant increase in CBF in both macro- and microcirculatory levels. We assume that the elevation of

epinephrine by stress causes an increase of CBF and changes in the tone of cerebral vessels that could be an initial factor triggering the BBB leakage. Indeed, 1h after music exposure, we have observed a decrease of the signal intensity from the TJ proteins, such as CLND5 and OCC, suggesting a temporal disorganization of the TJ assembly. It is important to note that already 4h after music effects, the signal intensity from the TJ proteins was within normal units that was preserved in the next day after the experiment. We hypothesize that these fast changes can be explained by the internalization of the TJ proteins resulting in a temporal loss of their surface in the space between endothelial cells that can be one of the mechanisms underlying music-induced OBBB. Our hypothesis is very close to the mechanism of BBB disruption via the internalization of VE-cadherin induced by VEGF [56]. **However, our findings cannot demonstrate the precise time of BBB recovery due to the high variability of BBB closing in different mice. It appears that the process of BBB recovery can start already during stress due to corticosterone-mediated reduce of BBB permeability [57]”.**

27. Akihiko U, Grubb J, Banks W, Sly W. 2007 Epinephrine enhances lysosomal enzyme delivery across the blood-brain barrier by up-regulation of the mannose 6-phosphate receptor. *Proc. Natl. Acad. Sci. USA*, **104**(31):12873- 8. (doi: 10.1073/pnas.0705611104)

28. Johansson B, Martinsson L. 1980 The blood-brain barrier in adrenaline-induced hypertension: circadian variations and modification by beta-adrenoreceptor antagonists. *Acta. Neurol. Scand*, **62**(2):96-102. (doi: 10.1111/j.1600-0404.1980.tb03009.x)

29. Murphy V, Johanson C. 1985 Adrenergic-induced enhancement of brain barrier system permeability to small nonelectrolytes: choroid plexus versus cerebral capillaries. *J. Cereb. Blood Flow Metab*, **5**(3):401-12. (doi: 10.1038/jcbfm.1985.55)

30. Sarmiento A, Borges N, Azevedo I. 1991 Adrenergic influences on the control of blood-brain barrier permeability. *Naunyn-Schmiedeberg's Archives of Pharmacology*. **343**: 633 – 637. (doi.org/10.1007/BF00184295)

55. Turner J, Parrish J, Hughes L, Toth L, Caspary D. 2005 Hearing in laboratory animals: strain differences and nonauditory effects of noise. *Comp Med*, **55**(1), 12–23. (<https://www.ncbi.nlm.nih.gov/pmc/articles/PMC3725606>)

56. Hebba J, Lecrair H, Azzi S, Roussel C, Scott M, Bidere N, Gavard J. 2013 The C-terminus region of β -arrestin-1 modulates VE-cadherin expression and endothelial cell permeability. *J. Cell. Commun. Signal*. **11**,37. (doi: 10.1186/1478-811X-11-37).

57. Sintona C.M., Fitcha T.E., Petty F., Haley R.W. 2000 Stressful Manipulations That Elevate Corticosterone Reduce Blood–Brain Barrier Permeability to Pyridostigmine in the Rat. *Pharmacology*. **165**(1) 99-105

Comment: Add to the manuscript that you used fluorescein isothiocyanate (FITC) to assess permeability to macromolecules.

Response: We added it.

In the second step, further optical measurement experiments employed **fluorescein isothiocyanate (FITC)-dextran 70 kDa (hereafter FITCD)** injected intravenously 1h after music exposure (the time of OBBB). The BBB disruption was determined both *ex vivo* using confocal microscopy on brain slices and *in vivo* using two-photon laser scanning microscopy (2PLSM) (Fig. 1, c-e). *Ex vivo* confocal microscopy revealed the accumulation of FITCD outside of brain capillaries 1h after music exposure. The leakage of FITCD was visualized by fluorescence outside the vessel walls (Fig. 1d and video 1 in SI). No leakage of FITCD was observed in the control group (no music) (Fig. 1 c and video 2 in SI). The leakage of FITCD in 2PLSM was observed also 1h after music exposure and was quantified by measuring the percentage fluorescence intensity of FITCD in the perivascular area (Fig. 1e). There were no any changes in the BBB permeability to FITCD 4h and 24h after music exposure.

Comment: Inconsistent usage of the abbreviation opening of BBB, sometimes it is OBBB and sometimes BBBO.

Response: We corrected in the whole text of manuscript.

Comment: In Fig.1 The Ki-maps from the same mice at different time points. Is it superimposed maps from all mice or one example mouse?

Response: Fig. 1f illustrates the changes of Ki values (arbitrary units) rate in the MRI signal intensity in 10 mice. Fig. 1g shows the Ki-maps from one mice at different time points. We modified subscription of Figure 1.

“Figures 1 f and g show the post music BBB disruption measured by magnetic resonance imaging (MRI) of gadolinium-diethylene-triamine-pentaacetic acid (Gd-DTPA) leakage. An analysis using rapid T1 weighting for Gd-DTPA is presented in Figure 1g. The image corresponding to 1 hour after sound is significantly brighter uniformly than others indicating leakage of the tracer molecule. This increase has been quantified by the rate of changes in the MRI signal intensity relates to the BBB permeability in the consequence from the 1st to 15th scanned images (Ki map) in Figure 1f, made for times 1, 4, & 24 hrs post music stimulus. The data given in Figure 1g indicate a statistically significant ($p < 0.001$) increase in the BBB permeability to Gd-DTPA in all regions of the brain 1h after music exposure. Note that the Ki map values reached 140 ± 3.7 arbitrary units ($p < 0.001$) only 1h after music exposure, while after 4h and 24h, changes in the Ki values were not observed, compared with the control group (no music)”.

Fig. 1. The *ex vivo* and *in vivo* results of music-OBBB (100 dB, 11-10,000 Hz, 2h intermittent mode: 60 sec – music; 60 sec – pause): a and b - *in vivo* real time fluorescent microscopy of the cerebral microvessels filled ED (no EB leakage) before music exposure (a) and EB extravasation from the cerebral capillaries into the brain tissues 1h after music impact indicating OBBB (b), $n=15$ in each group; c and d - Confocal imaging of brain slices demonstrating the BBB permeability to FITCD in mice subjected to loud music, where (c) – FITCD intravenous injection but no music exposure (FITCD is constrained to vessels) and d - FITCD injection 1h after music exposure, substantial leakage indicated by diffuse cloud around vessels, $n=10$ in each group; e - *In vivo* 2PLSM of the BBB permeability to FITCD in mice subjected to loud music expressed as the percentage fluorescence intensity in perivascular area. Data are presented as mean \pm SEM, $n = 10$, *** - $p < 0.01$; f and g – The MRI analysis of the BBB permeability to Gd-DTPA in mice subjected to loud music, where f - The Ki values show (arbitrary units) rate of changes in the MRI signal intensity, data are presented as mean \pm SEM, $n = 10$, *** $p < 0.01$ and g - The Ki-maps from one mice at different time points.

Comment: Page 9, line 5. "4h (histological data)". I believe it should be for 4 weeks.

Response: To analyze the short-term and delayed effects of loud music on the brain and auditory system injury, whole brain and the cochlear histologic examination was performed using haematoxylin and eosin staining and TUNEL staining for apoptosis 1h (the time of OBBB), 4h (the time of BBB closing), and 4 weeks (delayed effects) after music exposure. The timing of the experiments was dictated by the fact that sound damage may not appear immediately, but after a certain time. For example, peripheral synaptic connections of cochlear neurons are the most

vulnerable elements in the cochlea and, in the vast majority of cases, cochlear nerve fibers degenerate only long time after the sound trauma [31]. In animal models exposed to sound stress, hair cell loss can be widespread within hours [31-35], whereas the loss of the spiral ganglion cell (SGCs) is typically not detectable for several weeks to month after sound exposure [31,34,35]. The apoptosis evaluation after ultrasound-OBBB usually performed for several hours to 4-5 weeks after OBBB [36, 37]. Therefore, the time of analysis of SGCs and apoptosis was chosen 1h (the time of OBBB) and 4 weeks (delayed effects) after music exposure. **The histological analysis was performed in the control group (no music), 1h, 4h and 4 weeks after music exposure. The 4h after music exposure was selected to exclude the perivascular edema, which develops during the first hours after OBBB [38].**

31. Liberman M.C., Kujawa S.G. 2017 Cochlear synaptopathy in acquired sensorineural hearing loss: Manifestations and mechanisms. *Hear Res.* **349**, 138–147. (doi:10.1016/j.heares.2017.01.003)
32. Lawner B.E., Harding G.W., Bohne B.A. 1997 Time course of nerve-fiber regeneration in the noise-damaged mammalian cochlea. *Int J Dev Neurosci.* **15**(4–5), 601–617. (doi.org/10.1016/S0736-5748(96)00115-3)
33. Wang Y, Hirose K, Liberman M.C. 2002 Dynamics of noise-induced cellular injury and repair in the mouse cochlea. *J Assoc Res Otolaryngol.* **3**(3):248–268. (doi: 10.1007/s101620020028)
34. Webster D.B., Webster M. 1978 Cochlear nerve projections following organ of corti destruction. *Otolaryngol.* **86**(2):342–353. (doi.org/10.1177/019459987808600228)
35. Sugawara M, Corfas G, Liberman M.C. 2005 Influence of supporting cells on neuronal degeneration after hair cell loss. *J Assoc Res Otolaryngol,* **6**(2):136–147. (doi: 10.1007/s10162-004-5050-1)
36. McDannold N, Vykhodtseva N, Raymond S, Jolesz F.A, Hynynen K. 2005 MRI-guided targeted blood-brain barrier disruption with focused ultrasound: histological findings in rabbits. *Ultrasound Med Biol,* **31**(11):1527–1537. (doi: 10.1016/j.ultrasmedbio.2005.07.010)
37. Hynynen K, McDannold N, Vykhodtseva N, Raymond S, Weissleder R, Jolesz F.A., Sheikov N.S. 2006 Focal disruption of the blood–brain barrier due to 260-kHz ultrasound bursts: a method for molecular imaging and targeted drug delivery. *J Neurosurg,* **105**(3):445-454. (doi: 10.3171/jns.2006.105.3.445)
38. Semyachkina-Glushkovskaya O., Kurths J., Borisova E., Sokolovski S., Mantareva V., Angelov I., Shirokov A., Navolokin N., Shushunova N., Khorovodov A., Ulanova M., Sagatova M., Agranivich I., Sindeeva O., Gekalyuk A., Bodrova A., Rafailov E. 2017 Photodynamic opening of blood-brain barrier. *Biomed Opt Express.* **8**(11): 5040–5048.

Comment: Suppl material, table 1. Data for 90dB. n=11, legend said n = 10. Also, it is not a clear # of 4 animals. In 4 mice, BBB did not open after 4 hours or after 1 hour?

Response: It was mistake and we corrected it. The legend shows n=15 for the groups with music duration of 2h. Data for 90 dB included mice with OBBB (n=11) and mice, which did not demonstrate OBBB (n=4) 1h after music exposure.

Table 1 SI. The effect of loud music on the BBB permeability to EBAC (µg/g tissue)

Sound level (dB) and time elapsed after sound exposure (h)	Content of EBAC (µg/g tissue)			
	No music (the control group)	Music duration during 2 h (intermittent)	Music duration during 1 h (intermittent)	Music duration during 0.25 h (continues)
100 dB				
immediately		0.15±0.01		
¼ h		0.12±0.01	0.11±0.05	0.14±0.02
½ h		0.10±0.05	0.14±0.01	0.17±0.06
1 h		0.13±0.03	0.12±0.05	0.12±0.02
4 h		2.60±0.06 ***	0.12±0.03	0.17±0.01
24 h		0.19±0.03	0.15±0.07	0.15±0.05
		0.16±0.03	0.11±0.09	0.17±0.02

90 dB			
immediately	0.15±0.08	0.11±0.06	0.17±0.01
1 h	2.70±0.04 ***(n=11)	0.16±0.03	0.16±0.08
1 h	0.18±0.06 (n=4)#	0.14±0.03	0.14±0.05
4 h	0.15±0.03	0.17±0.02	0.18±0.01
24 h	0.19±0.07		
70 dB			
immediately	0.13±0.02	0.16±0.07	0.12±0.03
1 h	0.17±0.08	0.11±0.04	0.14±0.09
4 h	0.19±0.06	0.13±0.02	0.14±0.02
24 h	0.19±0.09	0.18±0.02	0.11±0.01

p<0.001: *** - vs. before music exposure (the control group), n=15 for the groups (music duration 2h) and n=10 for the groups (music duration 0.25-1 h); # - the number of mice without the BBB opening.

Comment: In supplemental material, add the description to videos.

Video resources:

Video 1 illustrates confocal imaging of music-induced opening of BBB to FITC Dextran 70 kDa (arrowed); FITC Dextran – Green (1 mg/25 g mouse, 0.5 % solution in saline, Sigma-Aldrich, i.v.); NG2 (pericyte marker) –Red: <https://youtu.be/Gk6uAMcCesg>

Video 2 illustrates confocal imaging of the cerebral microvessels in the control group (before music exposure; FITC Dextran – Green (1 mg/25 g mouse, 0.5 % solution in saline, Sigma-Aldrich, i.v.); NG2 (pericyte marker) –Red: <https://youtu.be/3CA1yNWdbCA>

Video 3 illustrates confocal imaging of the deep cervical lymph node after music-induced opening of BBB to FITC Dextran 70 kDa; FITC Dextran – Green (1 mg/25 g mouse, 0.5 % solution in saline, Sigma-Aldrich, i.v.), Lyve1/Prox1 (markers of the lymphatic endothelium) - Blue/Red: <https://youtu.be/PtZb-3nG45s>

Video 4 illustrates confocal imaging of the deep cervical lymph node in the control group (before music exposure); FITC Dextran – Green (1 mg/25 g mouse, 0.5 % solution in saline, Sigma-Aldrich, i.v.), Lyve1/Prox1 (markers of the lymphatic endothelium) - Blue/Red: <https://youtu.be/Yr8yglqMEzY>

Comment: Page 1. Correct typo in " All procedures were performed in accordance with the "Guide for the Care and Use of Laboratory Animals". The experimental protocols were approved by the Local Bioethics Commission of the Saratov State University (Protocol No. 7); the Institutional Animal Care and Use Committee of the University of New Mexico, USA (#200247).

Response: We corrected it.

All procedures were performed in accordance with the Guide for the Care and Use of Laboratory Animals. The experimental protocols were approved by the Local Bioethics Commission of the Saratov State University (Protocol No. 7) and the Institutional Animal Care and Use Committee of the University of New Mexico, USA (200247).

We would like to thanks again Reviewer for great help in improving of our manuscript. We really appreciate you taking the time out to share your experience with us.

Authors

Appendix E

Comment: Semyachkina-Glushkovskaya et al. provide a putative mechanistic explanation behind the use of rock music to affect blood-brain barrier permeability (BBB-p) in mice. They used Scorpions' "Still Loving You" at 70 dB, 90 dB, and 100dB at .25hr, 1hr, and 2hr time points, to determine BBB permeability to Evans Blue Albumin Complex (EBAC, 68.5 kDa) between 1-4hrs after exposure. This work is most innovative, and there is practically the only team that is addressing such questions. They have a history of rigorous research.

In and Ex vivo real-time fluorescent microscopy analysis confirmed highest concentrations of EBAC within cerebral microvessels after 1hr exposure to music (100 dB, 11-10,000 Hz, 2h intermittent mode: 60 sec – music; 60 sec – pause). A second analysis of BBB-p involved ex vivo confocal microscopy on brain slices and in vivo two-photon laser scanning microscopy (2PLSM). Ex vivo confocal microscop revealed the accumulation of FITCD outside of brain capillaries 1h after music exposure. No leakage of FITCD was observed in the control group (no music). The leakage of FITCD in 2PLSM was also observed 1h after music exposure and was quantified by measuring the percentage fluorescence intensity of FITCD in the perivascular area. A third analysis of BBB-p involved magnetic resonance imaging (MRI) of gadolinium-diethylene-triamine-pentaacetic acid (Gd-DTPA) leakage. Results indicate a statistically significant (p revealed the accumulation of FITCD outside of brain capillaries 1h after music exposure. No leakage of FITCD was observed in the control group (no music). The leakage of FITCD in 2PLSM was also observed 1h after music exposure and was quantified by measuring the percentage fluorescence intensity of FITCD in the perivascular area. A third analysis of BBB-p involved magnetic resonance imaging (MRI) of gadolinium-diethylene-triamine-pentaacetic acid (Gd-DTPA) leakage. Results indicate a statistically significant ($p<.001$) increase in the BBB permeability to Gd-DTPA in all regions of the brain 1h after music exposure. Altogether, the results demonstrate that loud music exposure (90-100 dB, 11 – 10.000 Hz, Scorpions "Still loving you") during 2 hrs in intermittent adaptive mode (1 min – sound; 1 min –pause) is accompanied by delayed (1h after music impact) and short-lasting (during 1-4 hrs) OBBB to low and high molecular weight compounds.

The stress response to loud music involved the release of epinephrine, as confirmed by an increase in the plasma level of epinephrine up to 3.1-fold (immediately after the music) vs. the normal state (9.0 ± 1.5 ng/ml vs. 2.9 ± 0.7 ng/ml, $p<.001$, $n=10$ in each group). One hour after music exposure (the time of OBBB), the level of epinephrine returned almost to the normal value of 3.9 ± 1.6 ng/ml ($n=10$) and was over the control units 4h after music impact (the time of BBB closing) (2.5 ± 0.1 ng/ml, $n=10$). To test this, they measured changes in cerebral blood flow at the venous (the Sagittal sinus) and microcirculatory levels in the same 10 mice before, immediately, 1h, 4h, and 24h after sound exposure. The results demonstrate that immediately after music rCBF was increased in both venous and microcirculatory levels compared with the control group (0.80 ± 0.03 a.u. vs. 0.58 ± 0.01 a.u., $p<.001$ for the cerebral microvessels; 1.22 ± 0.01 a.u. vs. 0.83 ± 0.02 a.u., $p,.001$ for the Sagittal sinus). One hour after sound exposure, when the BBB was opened, rCBF tended to decrease but continued to be higher than the normal level of rCBF (0.77 ± 0.08 a.u. vs. 0.58 ± 0.01 a.u., $p<.05$ for the cerebral microvessels; 0.92 ± 0.07 a.u. vs. 0.83 ± 0.02 a.u., $p<.05$ for the Sagittal sinus).

Finally, they studied integrity of tight junction proteins (claudin-5 (CLDN-5), occluding (OCC) and zonula occludens (ZO-1)immediately after music effects, during OBBB (1h after the music), and in the time of BBB recovery (4h and 24h after the music) compared with the control group (no music). Results show that the signal intensity from CLDN-5 and OCC but not from ZO-1 were significantly decreased in the time of OBBB. There were no changes in the TJ

assembly immediately after listening to music as well as in the time of BBB restoration (4h and 24 after the music). Importantly, music-OBBB for FITCD was accompanied by lymphatic clearance of FITCD from the brain with its accumulation in the enlarged lymphatic vessels of dcLNs (deep cervical lymph nodes) that was not observed in intact mice. Using confocal imaging of the deep cervical lymph nodes (dcLNs), which are the first anatomical station of the cerebral spinal fluid (CSF) outflow, they show that music-induced OBBB for FITCD was accompanied by FITCD lymphatic clearance from the brain with its accumulation in the enlarged lymphatic vessels of dcLNs ($22.3 \pm 1.5 \mu\text{m}$ vs. $37.3 \pm 2.0 \mu\text{m}$, $p < .001$) that was not observed in intact mice. The limitation section could be expanded. For example, variability in the potential effects of the complexity of the music used in this study may have skewed results. The musical selection used in this study was Scorpions' "Still Loving You." The authors tested varying loudness levels in decibels (70, 90, and 100), varying durations (.25hr, 1hr, and 2hr), and varying deliveries (60sec music – 60sec pause). Comparison/contrast of effect(s) upon BBB-p of other musical stimuli/selections was not conducted. Such a comparison/contrast may have concretely concluded that volume level is indeed the sole reason for BBB-p, rather than effects of other musical elements (e.g., melody, timbre, harmony, rhythm, tempo, texture, form, etc.), and/or combination of volume with other musical elements upon BBB-p. Inclusion of comparator musical piece(s)/song(s) involving distinctly different musical qualities to "Still Loving You" would provide complementary evidence of the efficacy of S-G et al. (2020) use of this song to draw their conclusions. Future work should include a piece(s) of the music of distinctly different tempo/rhythm, key signature/mode, instrumentation/timbre/arrangement, and/or texture (e.g., homophony, heterophony, etc.). Furthermore, instrumental vs. vocal vs. instrumental/vocal combinations could be tested to concretely exclude these variables.

Response: We want to express our sincere gratitude for hard work with our manuscript and deep analysis of our results as well as for constructive and helpful advices. We added the limitations of our study in conclusion.

“Conclusion

This pioneering experimental study has discovered the important phenomenon of OBBB in healthy mouse brain induced by loud rock music (90-100 dB, 11 – 10.000 Hz, Scorpions "Still loving you"). The listening of loud music during 2h in intermittent adaptive mode is accompanied by delayed (1h after music exposure) and short-lasting (during 1-4 hrs) OBBB for high and low molecular weight molecules. We assume that an elevation of epinephrine by sound stress causes an increase of CBF and changes in the tone of cerebral vessels that could be an initial factor triggering the BBB leakage associated with a decrease in expression of TJ proteins. The music-induced OBBB is reversible with quickly restoration of the BBB function without the injuries of brain tissues and the cochlea. It can be related to the repetitive mode of music exposure and post-OBBB activation of lymphatic clearance of molecules penetrating into the brain via OBBB.

Our results stimulate a revision of the classic knowledge about the BBB nature. The BBB opens itself in a healthy brain as a response to sound stress. Despite the fact that we did not find any injuries of the brain tissues and the cochlea after loud music-OBBB due to limitation of our studies of delayed music effects by 4 weeks, even temporal OBBB can be harmful for the brain via opening the door for viruses, bacteria and toxins. It is important to note that millions of teenagers and professional rock, jazz and symphony orchestra musicians are in a potential risk

group of OBBB due to exposure to loud music in concerts, nightclubs, discotheques, and pubs. Therefore, more detailed studies of loud music-induced OBBB are required for further investigations in human.

We also believe that our findings can be the basis for novel strategies in optimization of sound-mediated methods for brain drug delivery. Despite the fact that music opens BBB in a non-targeted manner, loud music has a high potential for clinical applications as an easy used, non-invasive, low cost, labeling free, perspective and completely new approach for the treatment of neurodegenerative disorders, such as Alzheimer's disease or myotrophic lateral sclerosis, which are associated with injuries of many brain regions and diffusive progression without specificity of the brain areas. We address to further research of music-induced OBBB as a proof of concept for exploration of sound-induced drug delivery to the brain. Such issues as the study of efficacy of the drug delivery into the brain and the effect of different levels of sound on the pharmacokinetics of the delivered drug can shed light on the development of a new alternative method of drug brain delivery for different neurodegenerative diseases, brain oncology and brain trauma.

This pilot study has limitations including using only one music composition, such as Scorpions' "Still Loving You", which we tested in varying loudness intensities (70 dB, 90dB, and 100dB), durations ($\frac{1}{4}$ h, $\frac{1}{2}$ h, 25hr, 1hr, and 2hr after music exposure) and deliveries (60sec music – 60sec pause). In further experiments, the effects of other musical stimulus on BBB will be studied focusing on the analysis of effects of musical elements in combination/comparison/contrast of timbre, harmony, rhythm, tempo, texture, homophony, heterophony using instrumental/vocal combinations for better understanding variability of complex of music effects on the BBB permeability".

We would like to thanks again Reviewer for great help in improving of our manuscript. We really appreciate you taking the time out to share your experience with us.

Authors